# Polynomial-Time Computation of Exact $\Phi$-Equilibria in Polyhedral Games

**Gabriele Farina**
MIT
gfarina@mit.edu

**Charilaos Pipis**
MIT
chpipis@mit.edu

## Abstract

It is a well-known fact that correlated equilibria can be computed in polynomial time in a large class of concisely represented games using the celebrated Ellipsoid Against Hope algorithm (Papadimitriou and Roughgarden, 2008; Jiang and Leyton-Brown, 2015). However, the landscape of efficiently computable equilibria in sequential (extensive-form) games remains unknown. The Ellipsoid Against Hope does not apply directly to these games, because they do not have the required "polynomial type" property. Despite this barrier, Huang and von Stengel (2008) altered the algorithm to compute exact extensive-form correlated equilibria.

In this paper, we generalize the Ellipsoid Against Hope and develop a simple algorithmic framework for efficiently computing saddle-points in bilinear zero-sum games, even when one of the dimensions is exponentially large. Moreover, the framework only requires a "good-enough-response" oracle, which is a weakened notion of a best-response oracle.

Using this machinery, we develop a general algorithmic framework for computing exact linear $\Phi$-equilibria in any polyhedral game (under mild assumptions), including correlated equilibria in normal-form games, and extensive-form correlated equilibria in extensive-form games. This enables us to give the first polynomial-time algorithm for computing exact linear-deviation correlated equilibria in extensive-form games, thus resolving an open question by Farina and Pipis (2023). Furthermore, even for the cases for which a polynomial time algorithm for exact equilibria was already known, our framework provides a conceptually simpler solution.

## 1   Introduction

The correlated equilibrium (CE), introduced by Aumann (1974), is one of the most seminal solution concepts in multi-player games. Contrary to the Nash equilibrium, in a correlated equilibrium the players' strategies are correlated by a fictitious *mediator* that can recommend (but not enforce) behavior. It is then up to this mediator to ensure that the distribution of recommendations does not incentivize any player to *deviate* from their recommended strategy. It is known that this type of equilibrium naturally emerges from the repeated interaction of learning agents (Hart and Mas-Colell, 2000). In practice, this means that one can compute $\epsilon$-approximate CEs in normal-form games by implementing suitable decentralized no-regret dynamics, of which several efficient implementations are known (see, *e.g.*, Blum and Mansour (2007) and Anagnostides et al. (2022)). However, this approach requires $\Omega(\mathrm{poly}(1/\epsilon))$ iterations to compute an $\epsilon$-approximate equilibrium, making it a non-viable choice for high-precision equilibrium computation. In a celebrated result, Papadimitriou and Roughgarden (2008) (with later refinements by Jiang and Leyton-Brown (2015)) devised an algorithm, called *Ellipsoid Against Hope*, that can compute an exact CE in a concisely represented normal-form game in polynomial time in the representation of the game. Their algorithm is an algorithmic version of the clever reduction of Hart and Schmeidler (1989) that casts the computation of CEs as a two-player zero-sum game.

38th Conference on Neural Information Processing Systems (NeurIPS 2024).

The positive results for normal-form games, however, do not transfer directly to the significantly more involved setting of *extensive-form games*. Extensive-form games are games played on a game tree and can model sequential and simultaneous moves, as well as imperfect information. Despite a significant stream of *positive* results related to learning and equilibrium computation in extensive-form games, the complexity of computing CEs in extensive-form games remains to this day a major open question (Farina and Pipis, 2023; von Stengel and Forges, 2008). Due to its conjectured intractability, researchers have resorted to considering the computation of weaker and generalized notions of correlated equilibrium. A key reference point in this space is given by the framework of Gordon et al. (2008), who define a generalized notion of CE called $\Phi$-equilibria. In a $\Phi$-equilibrium, every player $p$ is endowed with a set $\Phi_p$ of *behavior transformation functions*. The goal of the fictitious mediator is then simply to recommend strategies such that no player could unilaterally benefit from deviating by using any of the functions $\phi \in \Phi_p$. In this language, a CE corresponds to the $\Phi$-equilibrium in which each $\Phi_p$ is the set of all possible functions from the strategy set of the player to itself. However, by considering appropriate subsets of behavior transformations, weaker supersets of CEs can be efficiently computed and learned through uncoupled learning dynamics. Notable examples of such equilibria in extensive-form games include the extensive-form correlated equilibrium (EFCE) (von Stengel and Forges, 2008), the extensive-form coarse-correlated equilibrium (EFCCE) (Farina et al., 2020), the normal-form coarse-correlated equilibrium (NFCCE) (Moulin and Vial, 1978), the recently-introduced linear-deviation correlated equilibrium (LCE) (Farina and Pipis, 2023), and others (Morrill et al., 2021).

Huang and von Stengel (2008) proposed a specialization of the Ellipsoid Against Hope algorithm to compute exact EFCE in extensive-form games. Later, Farina et al. (2022a) showed efficient no-regret dynamics that converge to the EFCE. More recently, there has been increased interest in understanding what is the $\Phi$-equilibrium that is the closest to CE while still enabling efficient computation and learning. Farina and Pipis (2023) introduced the linear-deviation correlated equilibrium (LCE) that arises from the set $\Phi_{\text{LIN}}$ of all linear-swap deviations in sequence-form strategies and devise efficient no-linear-swap regret dynamics to approximate it. The LCE captures all notable notions of equilibrium that were previously known to be efficiently computable (including EFCE, EFCCE, and NFCCE). However, Farina and Pipis (2023) left open the key question as to whether LCEs themselves can also be computed exactly in polynomial time, akin to the generalization of the Ellipsoid Against Hope algorithm by Huang and von Stengel (2008), as opposed to just learned via uncoupled learning dynamics. The approach by Huang and von Stengel (2008) relies heavily on the combinatorial structure of the deviation functions that define EFCE, resulting in a rather involved algorithm. This is in stark contrast to the simple framework for constructing $\Phi$-regret minimizers championed by Gordon et al. (2008). This begs the natural question:

> *Can we always construct an efficient algorithm for exactly computing $\Phi$-equilibria, when there exists an efficient no-$\Phi$-regret minimizer?*

In other words, can we create a simple and general framework in the spirit of Gordon et al. (2008) that can enable us to construct algorithms for the exact computation of $\Phi$-equilibria in polyhedral games for any $\Phi \subseteq \Phi_{\text{LIN}}$? We answer this question in the affirmative.

**Contributions.** In this paper, we propose a framework for computing exact $\Phi$-equilibria in general polyhedral games. Our framework recovers all positive results established by Papadimitriou and Roughgarden (2008), and crucially applies to polyhedral games such as extensive-form games. Using our framework, we develop the first polynomial-time algorithm for computing exact linear-deviation correlated equilibria in extensive-form games, thus resolving an open question by Farina and Pipis (2023). Furthermore, even for the cases for which a polynomial time algorithm for exact equilibria was already known (CEs in normal-form games (Papadimitriou and Roughgarden, 2008; Jiang and Leyton-Brown, 2015) and EFCEs in extensive-form games (Huang and von Stengel, 2008)), our framework provides a conceptually simpler solution.

We show that to compute an exact $\Phi$-equilibrium in a polyhedral game, the following three conditions are sufficient:

1. The game satisfies the "polynomial utility gradient property" (Assumption 4.2) which states that given a product distribution over the joint strategy space of all $n$ players, we can efficiently compute the expectation of the gradient of any player's utility. This is a natural generalization of the "polynomial expectation property" of Papadimitriou and Roughgarden

(2008), and it is a rather low bar to clear (in fact, it is implicitly assumed in every no-regret learning algorithm).

2. $\Phi$ is a set of *linear* transformations (*i.e.*, of the form $\phi(\boldsymbol{x}) = \mathbf{B}\boldsymbol{x}$ for some matrix $\mathbf{B}$) that map the strategy set to itself. This is a technical requirement so that the expectation operator and the application of the deviation function can commute.[1] This condition is satisfied by all notions of $\Phi$-equilibrium mentioned above, including EFCE and LCE in extensive-form games, and CE in normal-form games.

3. The set $\Phi$ of transformations is a polytope that admits a polynomial-time separation oracle.

The separation oracle requirement in the third condition is known to be equivalent to efficient linear optimization (Grötschel et al., 1993). In essence, this means that giving a polynomially-sized characterization of a set $\Phi$ of linear transformations for a polyhedral set is a sufficient condition to provide both efficient no-regret learning dynamics and an efficient algorithm for computing exact $\Phi$-equilibria. This is exactly what we achieve by applying our result to the set of linear-swap deviations that was recently characterized (Farina and Pipis, 2023; Zhang et al., 2024b) as a polytope of polynomially many constraints and was used to prove efficient no-linear-swap regret dynamics. In light of all these considerations, our framework can be thought of as the counterpart of the $\Phi$-regret minimization framework by Gordon et al. (2008), but for computation of *exact* equilibria rather than regret minimization.

At the heart of our construction, our main technical tool is a generalization of the methodology of the Ellipsoid Against Hope (Papadimitriou and Roughgarden, 2008; Jiang and Leyton-Brown, 2015) to general polyhedral bilinear games. In more detail, we give a new constructive proof of the minimax theorem for players with polyhedral strategy sets, by using only a weakened type of a best-response oracle that we coin Good-Enough-Response (GER) oracle. An interesting property of the GER oracle is that it can be computationally tractable even when the respective best-response oracle is intractable, as we show in Section 4. This algorithmic idea is likely of independent interest and is especially useful when the strategy space of one of the players is very large but there exists an efficient GER oracle that outputs sparse solutions (*e.g.*, vertices of a high-dimensional polytope). This is exactly the type of problem we face when we need to compute exact $\Phi$-equilibria in polyhedral games and we then proceed to apply this machinery to the above question. Interestingly, in order to show the existence of structured good-enough responses in the context of $\Phi$-equilibria, we use an argument based on the existence of an efficient fixed-point oracle for each deviation $\phi \in \Phi$. Such an ingredient was fundamental (albeit used differently; see Hazan and Kale (2007) for a discussion of the role played by fixed-point oracles in the construction of no-$\Phi$-regret algorithms) also in Gordon et al. (2008). In our case, it is one of the technical insights that enable us to sidestep much of the intricacy encountered by Huang and von Stengel (2008).

We defer all proofs of the paper to the appendix.

**Related work.** We include an extensive discussion of related work in Appendix A.

## 2   Preliminaries

In this section, we introduce some basic concepts and definitions that will be used in developing our framework.

### 2.1   Polyhedra, polytopes, and convex sets

**Definition 2.1** (Rational polyhedron). *A rational polyhedron* $\mathcal{P} = \{\boldsymbol{x} \in \mathbb{R}^n \mid \mathbf{A}\boldsymbol{x} \leq \boldsymbol{b}\}$ *is the solution set of a system of linear inequalities with rational coefficients. We say that* $\mathcal{P}$ *has **facet-complexity** $\varphi$ if there exists a system of linear inequalities, where each inequality has encoding length* [2] *at most $\varphi$, and whose solution set is $\mathcal{P}$. A rational polyhedron that is* bounded *is called a* rational polytope.

---

[1]Note that going beyond linear transformations can introduce several complications. Most notably, in a recent paper, Zhang et al. (2024a) observe that computing exact fixed-points of non-linear transformations might be PPAD-hard and they instead introduce a new way to perform regret minimization using "approximate expected fixed-points".

[2]The encoding length of an inequality is the total amount of bits required to represent all of its coefficients.

One important property of rational polytopes that we will use repeatedly throughout the paper is that they can equivalently be written as the convex hull of a *finite* number of points. We call these points the *vertices* $V(\mathcal{P})$ of polytope $\mathcal{P}$. Additionally, the vertices of a rational polytope always have rational coordinates and encoding length $\text{poly}(\varphi)$ (Grötschel et al., 1993, Lemma 6.2.4).

Since we are interested in constructing algorithms that perform exact computations, any discussion of non-rational numbers is not relevant. Thus, from now on, every time we deal with a polytope we will mean a rational polytope.

In our algorithm, we will also make use of the concept of *conic hull*, which is introduced next.

**Definition 2.2** (Conic hull). *The conic hull of a convex set $\mathcal{X}$ is $\mathbb{R}_+\mathcal{X} = \{t \cdot \boldsymbol{x} \mid t \geq 0, \boldsymbol{x} \in \mathcal{X}\}$. Furthermore, if $\mathcal{X}$ is a rational polyhedron, its conic hull is also a rational polyhedron.*

## 2.2 Game theory definitions

We begin by defining polyhedral games, following Gordon et al. (2008). But first, we need to define multi-linear functions.

**Definition 2.3** (Multi-linear function). *Let $V_1, \ldots, V_n$ be vector spaces. A function $f : V_1 \times \cdots \times V_n \to \mathbb{R}$ is said to be multi-linear if for each $p \in [n]$ and fixed $\boldsymbol{v}_{-p} \in V_{-p}$ the function $f(\boldsymbol{v}_p, \boldsymbol{v}_{-p})$ is linear in $\boldsymbol{v}_p \in V_p$. In other words, if $\nabla f(\boldsymbol{v}_{-p})$ is the gradient of $f(\boldsymbol{v}_p, \boldsymbol{v}_{-p})$ with respect to $\boldsymbol{v}_p$ when $\boldsymbol{v}_{-p}$ is fixed, then $f(\boldsymbol{v}_p, \boldsymbol{v}_{-p}) = \boldsymbol{v}_p \cdot \nabla f(\boldsymbol{v}_{-p})$.*

**Definition 2.4** (Polyhedral game). *In a polyhedral game with $n$ players, every player $p \in [n]$ has a polytope[3] strategy set $\mathcal{A}_p \subset \mathbb{R}^{d_p}$ and a multi-linear utility function $u_p : \mathcal{A}_1 \times \cdots \times \mathcal{A}_n \to \mathbb{R}$*

Some notable examples of polyhedral games are: normal-form games, where every player has a probability simplex as their strategy set, and extensive-form games, where the strategy sets of the players are the sets of sequence-form strategies (Romanovskii, 1962; Koller et al., 1996; von Stengel, 1996). We will refer to the encoding length of the game as the *size of the game*. In games of interest this is usually much smaller than holding the full utility function; for example, extensive-form games are encoded using a game tree and different classes of normal-form games can have other succinct descriptions (Papadimitriou and Roughgarden, 2008).

A sub-class of polyhedral games that will be particularly useful in our paper is that of bilinear zero-sum games, which is defined below.

**Definition 2.5** (Bilinear zero-sum game). *Let $\mathcal{X} \subset \mathbb{R}^M$, $\mathcal{Y} \subset \mathbb{R}^N$ be two rational polytopes. A bilinear zero-sum game is a game between two players with strategy sets $\mathcal{X}$ and $\mathcal{Y}$ such that the utility of the $\mathcal{X}$-player is $u_1(\boldsymbol{x}, \boldsymbol{y}) = \boldsymbol{x}^\top \mathbf{A} \boldsymbol{y}$, for some $\mathbf{A} \in \mathbb{Q}^{M \times N}$, and the utility of the $\mathcal{Y}$-player is $u_2(\boldsymbol{x}, \boldsymbol{y}) = -u_1(\boldsymbol{x}, \boldsymbol{y})$*

We can now define the notion of a $\Phi$-equilibrium, which generalizes the correlated equilibrium for arbitrary $n$-player polyhedral games and sets of strategy transformations $\Phi$. Before we do that, we first need to define the *corner game* $\Gamma(G)$ of a polyhedral game $G$, following Gordon et al. (2008); Marks (2008). This is the game that arises if we let the action sets of every player $p$ be equal to the vertices $V(\mathcal{A}_p)$ of the polytope strategy set of that player. Note that since $\mathcal{A}_p$ is a polytope, it will have a finite number of vertices. The utilities of this game for a player $p \in [n]$ and pure strategy profile $\boldsymbol{s} \in V(\mathcal{A}_1) \times \cdots \times V(\mathcal{A}_n)$ are simply given by $u_p(\boldsymbol{s})$. In this paper, we will denote the vertices of every strategy set in a polyhedral game as $\Pi_p = V(\mathcal{A}_p)$. We are now ready to define the $\Phi$-equilibrium.

**Definition 2.6** ($\Phi$-equilibrium). *Let $G$ be a polyhedral game of $n$ players and $\Phi_p$ be a set of strategy transformations $\phi_p : \mathcal{A}_p \to \mathcal{A}_p$ for each player $p \in [n]$. A $\{\Phi_p\}$-equilibrium for $G$ is a joint distribution $\mu \in \Delta(\Pi_1 \times \cdots \times \Pi_n)$ on the pure strategy profiles of $\Gamma(G)$, such that for every player $p \in [n]$ and deviation $\phi \in \Phi_p$ it holds*

$$\mathbb{E}_{\boldsymbol{s} \sim \mu}[u_p(\boldsymbol{s})] \geq \mathbb{E}_{\boldsymbol{s} \sim \mu}[u_p(\phi(\boldsymbol{s}_p), \boldsymbol{s}_{-p})].$$

*That is, no player $p$ has an incentive to unilaterally deviate from the recommended joint strategy $\boldsymbol{s}$ using any transformation $\phi \in \Phi_p$.*

---

[3]Despite their name, polyhedral games have strategy sets that are *polytopes*, that is, *bounded* polyhedra.

# 3 A simple framework for computing equilibria in bilinear zero-sum games using good-enough-response (GER) oracles

We begin by introducing a simple algorithmic framework (Theorem 3.1) for computing min-max equilibria in bilinear zero-sum games. As mentioned before, it relies on the idea of good-enough-responses. The motivation behind this is that sometimes a best-response oracle is not known, or even NP-hard to construct (as we prove in Theorem 4.7). On the contrary, good-enough-responses might be a readily available primitive. For example, we will see in Section 4 that a good-enough-response oracle materializes through the use of fixed-point oracles for transformations $\phi \in \Phi$ and this enables us to devise polynomial time algorithms for computing exact $\Phi$-equilibria in polyhedral games.

Let us assume that we have a bilinear zero-sum game $\mathcal{G}(\mathcal{X}, \mathcal{Y}, \mathbf{A})$, where the strategy sets $\mathcal{X} \subset \mathbb{R}^M, \mathcal{Y} \subset \mathbb{R}^N$ are rational polytopes. We typically assume that $M \gg N$. Additionally, let

$$\mathtt{OPT} = \max_{\boldsymbol{x} \in \mathcal{X}} \min_{\boldsymbol{y} \in \mathcal{Y}} \boldsymbol{x}^\top \mathbf{A} \boldsymbol{y},$$

be the value of the game at equilibrium, which is known to us. *In the rest of the paper we assume that $OPT = 0$.* This is without loss of generality because otherwise, it is possible to create a new game with this property by augmenting the vectors $\boldsymbol{x}, \boldsymbol{y}$ with an extra dimension as follows:

$$\begin{bmatrix} \boldsymbol{x}^\top & 1 \end{bmatrix} \begin{bmatrix} \mathbf{A} & \mathbf{0} \\ \mathbf{0}^\top & -\mathtt{OPT} \end{bmatrix} \begin{bmatrix} \boldsymbol{y} \\ 1 \end{bmatrix} = \boldsymbol{x}^\top \mathbf{A} \boldsymbol{y} - \mathtt{OPT}.$$

Our framework is a formalization of the following observation. The statement

(S1) Given any $\boldsymbol{y} \in \mathcal{Y}$ we can find some $\boldsymbol{x} = \boldsymbol{x}(\boldsymbol{y}) \in \mathcal{X}$ such that $\boldsymbol{x}^\top \mathbf{A} \boldsymbol{y} \geq 0$.

implies the following

(S2) There exists $\boldsymbol{x}^* \in \mathcal{X}$ such that $(\boldsymbol{x}^*)^\top \mathbf{A} \boldsymbol{y} \geq 0$ for all $\boldsymbol{y} \in \mathcal{Y}$.

This follows from the minimax theorem (Neumann, 1928), as the first statement (S1) is equivalent to $\min_{\boldsymbol{y}} \max_{\boldsymbol{x}} \boldsymbol{x}^\top \mathbf{A} \boldsymbol{y} \geq 0$, while the second statement (S2) is equivalent $\max_{\boldsymbol{x}} \min_{\boldsymbol{y}} \boldsymbol{x}^\top \mathbf{A} \boldsymbol{y} \geq 0$.

We are interested in the following question: *"Is there an efficient algorithm that when given access to an oracle for (S1), it constructs a solution $\boldsymbol{x}^*$ for (S2) represented as a mixture of a small number of oracle responses?".*

## 3.1 Good-Enough-Response (GER) oracle

We begin by formally defining the oracle we presented previously, which we coin a Good-Enough-Response (GER) oracle. It is defined as follows:

```
GER(y):
    return (x, x⊤A) ∈ X × ℚ^N  s.t.   x⊤Ay ≥ OPT = 0
```

where $\boldsymbol{y} \in \mathcal{Y} \subset \mathbb{R}^N$, and $\mathtt{OPT} = 0$ as was discussed earlier. Note that this is not a best-response oracle, because it does not return an $\boldsymbol{x} \in \mathcal{X}$ that maximizes the utility of the max-player. Rather, it suffices to return a "good enough response", hence the name.

In fact, our algorithms will often need to query a GER oracle for $\boldsymbol{y}' \in \mathbb{R}_+ \mathcal{Y}$ and not just for vectors in $\mathcal{Y}$. This however is not a problem because it suffices to find any $\boldsymbol{y} = \boldsymbol{y}'/\alpha$ for some $\alpha > 0$ and $\boldsymbol{y} \in \mathcal{Y}$ and then query $\mathtt{GER}(\boldsymbol{y})$ instead. To find such a $\boldsymbol{y}$ efficiently we can again, without loss of generality, assume that all vectors $\boldsymbol{y} \in \mathcal{Y}$ are augmented with an extra dimension (call it $\boldsymbol{y}[\varnothing]$) such that $\boldsymbol{y}[\varnothing] = 1$ for all $\boldsymbol{y} \in \mathcal{Y}$. Then we can find the desired scaling factor immediately because $\boldsymbol{y}'[\varnothing] = \alpha$ if and only if $\boldsymbol{y}' = \alpha \boldsymbol{y}$ for $\boldsymbol{y} \in \mathcal{Y}$.

In addition to a good-enough-response oracle, our algorithm also requires a separation oracle $\mathtt{SEP}_\mathcal{Y}$ for the polytope $\mathcal{Y}$, which can be easily converted to a separation oracle for $\mathbb{R}_+ \mathcal{Y}$ by the same "augmenting" argument as before. Combining these two, we can make the final separation oracle (Algorithm 2) that is needed to execute the ellipsoid method on $(D)$, as presented later. Specifically, if $\boldsymbol{y} \notin \mathbb{R}_+ \mathcal{Y}$ then we simply return a separating hyperplane via $\mathtt{SEP}_{\mathbb{R}_+ \mathcal{Y}}$, else we return a good-enough-response from $\mathtt{GER}$.

## 3.2 The framework

Our goal is to compute an $x \in \mathcal{X}$ that is an optimal (min-max) strategy for the max-player. Equivalently, we seek to find a solution to the following linear program

$$\text{find } x \in \mathcal{X} \qquad \text{s.t. } \min_{y \in \mathcal{Y}} x^\top \mathbf{A} y \geq 0 \qquad\qquad (P)$$

This is an LP with $M$ variables which is typically assumed to be much greater (even super-exponentially greater) than $N$. When faced with this situation, one might want to attempt to directly solve the dual of $(P)$ using the ellipsoid method. However, this would require a proper separation oracle for the dual problem, which corresponds to a linear optimization oracle, or at least a best-response oracle. But as we explained, the oracle access we have is weaker.

Instead, we focus on the below linear program. Note that for any $y \in \mathbb{R}_+ \mathcal{Y}$, `GER(y)` should always return an $x \in \mathcal{X}$ such that $(x^\top \mathbf{A}) y \geq 0$, which is a violated constraint of $(D)$. Thus, we can combine `GER` and a separation oracle for $\mathbb{R}_+ \mathcal{Y}$ (as in Algorithm 2) to make a separation oracle for this LP.

$$\text{find } y \in \mathbb{R}_+ \mathcal{Y} \qquad \text{s.t. } \max_{x \in \mathcal{X}} x^\top \mathbf{A} y \leq -1 \qquad\qquad (D)$$

By the Generalized Farkas lemma (Lemma B.2) and the fact that $(P)$ is feasible, it immediately follows that $(D)$ must be infeasible. Despite the infeasibility, and following the "Against Hope" step of Papadimitriou and Roughgarden (2008), we execute the ellipsoid method on $(D)$ using Algorithm 2 as a separation oracle. The ellipsoid method will run for a number $L = \text{poly}(N)$ of steps and then conclude that $(D)$ is infeasible. Let $x_1, \ldots, x_L$ be the response vectors returned by `GER` during this process. We now consider a "compressed" version of the previous LP that only uses vectors $x$ from the convex hull $\text{co}\{x_k\}$ of these responses.

$$\text{find } y \in \mathbb{R}_+ \mathcal{Y} \qquad \text{s.t. } \max_{x \in \text{co}\{x_k\}} x^\top \mathbf{A} y \leq -1 \qquad\qquad (D')$$

We argue that this LP must also be infeasible; the ellipsoid method is a deterministic algorithm and if we execute it on $(D')$ it will go through the same sequence of candidate points $y_k$, to which we can respond with the same sequence of separating hyperplanes as before. These hyperplanes will still be valid for $(D')$ because all of the response vectors we used previously exist in $\text{co}\{x_k\}$.

Now, using Generalized Farkas lemma (Lemma B.2) again and the fact that $(D')$ is infeasible, it follows that the LP shown below must be feasible.

$$\text{find } x \in \text{co}\{x_k\} \qquad \text{s.t. } \min_{y \in \mathcal{Y}} x^\top \mathbf{A} y \geq 0 \qquad\qquad (P')$$

This is a "compressed" version of $(P)$, because now every vector $x \in \text{co}\{x_k\}$ can be represented as a vector of size $L$ that corresponds to a convex combination of the response vectors $x_1, \ldots, x_L$. Finally, since $(P')$ is an LP with only a polynomial number of variables, we can solve it in polynomial time using any LP algorithm. This will clearly be a feasible solution for our initial LP $(P)$, because $\text{co}\{x_k\} \subset \mathcal{X}$. The full algorithm is shown below, in Algorithm 1. Note that in reality we only use the LPs $(D)$ and $(P')$. The rest were used as intermediate steps for the presentation of the algorithm.

---

**Algorithm 1:** Ellipsoid Against Hope for bilinear zero-sum games

---

**Input:** Separation oracle $\text{SEP}_{\mathbb{R}_+ \mathcal{Y}}$ for $\mathbb{R}_+ \mathcal{Y}$, and a good-enough-response oracle `GER`.
**Output:** A sparse solution $x^*$ of $(P)$ represented as a mixture of `GER` oracle responses.
Execute the ellipsoid method on $(D)$, using Algorithm 2 as a separation oracle;
Create $(P')$ using the response vectors and compute a feasible solution $x^*$;

---

**Theorem 3.1.** *If the following hold*

1. *$\mathcal{X} \subset \mathbb{R}^M, \mathcal{Y} \subset \mathbb{R}^N$ are rational polytopes and $\mathcal{Y}$ has facet-complexity at most $\varphi$,*

2. *we have access to a separation oracle $\text{SEP}_{\mathcal{Y}}$ for $\mathcal{Y}$ and a good-enough-response oracle `GER`,*

3. *the encoding length of $x^\top \mathbf{A}$ is at most $\varphi$ for all `GER` oracle responses and all vertices of $\mathcal{X}$,*

*then Algorithm 1 runs in $\text{poly}(N, \varphi)$ time, performs $L = \text{poly}(N, \varphi)$ oracle calls, and computes an exact solution $x^*$ of $(P)$ that is a mixture of at most $N$ oracle responses. In particular, the encoding length of $x^*$ depends polynomially on the encoding length of the `GER` oracle responses.*

Note that since we have assumed that $M \gg N$, it would not make sense for the final solution $\boldsymbol{x}^*$ to have encoding length $\mathrm{poly}(M)$, as this would invalidate the whole algorithm. In order for the solution to make sense, the GER oracle must only give responses with low encoding length. This is exactly the case in Section 4, where $M$ is a doubly-exponential quantity in the size of the problem, while the GER responses are vectors with only one non-zero entry.

## 4 Computing linear $\Phi$-equilibria in polynomial time

We have seen in Section 3 how one can compute exact min-max equilibria using good-enough-response (GER) oracles. Now it is time to apply this machinery in the problem of computing *exact* $\Phi$-equilibria in polyhedral games. Crucially, the factor that enables us to utilize the framework of Section 3 is the existence of an efficient GER oracle, which effectively boils down to constructing a product distribution consisting of fixed-points for the strategies of every player of the game.

Let $G$ be any polyhedral game (Definition 2.4) with $n$ players and strategy sets $\mathcal{A}_p \subset \mathbb{R}^{d_p}$ for $p \in [n]$. In this section we apply the framework we developed previously to construct an algorithm that computes an exact $\Phi$-equilibrium of $G$ in polynomial time when $\Phi$ is a polytope containing valid linear transformations from polyhedral strategies to polyhedral strategies. Notable examples of sets with these properties are the trigger deviations used for EFCE (Farina et al., 2022a), and the linear-swap deviations used for LCE (Farina and Pipis, 2023) in extensive-form games.

The general idea of our construction is that of the existence proof by Hart and Schmeidler (1989) that casts the problem of $\Phi$-equilibrium computation as one of computing a min-max equilibrium in a two-player zero-sum meta-game between a "Correlator", who acts upon the simplex of all pure strategy profiles, and a "Deviator", whose actions correspond to deviations for every player. We call this a *Correlator-Deviator game*.

To make this idea applicable to polyhedral games, we generalize it as follows. We define a bilinear zero-sum meta-game with strategy sets $\mathcal{X}, \mathcal{Y}$ for the two players, where $\mathcal{X}$ is the set of all joint distributions over strategy profiles, $\mathcal{X} = \Delta(\Pi_1 \times \cdots \times \Pi_n)$ (hence, a polytope) and $\mathcal{Y}$ is the Cartesian product of $\Phi_p$ for all players $p$, $\mathcal{Y} = \Phi_1 \times \cdots \times \Phi_n$, which is a convex set – and in our case, a polytope.

We remark here that linear transformations $\phi_p$ can be represented using a matrix $\mathbf{B}_p$ such that $\phi_p(\boldsymbol{x}_p) = \mathbf{B}_p \boldsymbol{x}_p$. Thus, when we say that $\Phi_p$ is a polytope, it means that there exists a system of inequalities that can describe the entries of the corresponding matrix $\mathbf{B}_p$ for every $\phi_p \in \Phi_p$. For notational convenience, we will interchangeably use $\Phi_p$ to denote either the set of transformation functions, or a polytope describing the vectors (flattened $\mathbf{B}_p$ matrices) that correspond to transformations. In any event, it should not matter which of the two representations we have, because they are completely equivalent.

The utility matrix $\mathbf{U}$ of the Correlator in the meta-game is shown below. Specifically, it has one row for each pure strategy profile $\boldsymbol{s} \in \Pi_1 \times \cdots \times \Pi_n$, and one column for each tuple $j = (p, a, b)$, where $a, b \in [d_p]$ are used as indices over strategy vectors $\boldsymbol{s}_p \in \mathcal{A}_p$. Additionally, we always want the final expression to have a quantity $(\sum_p \mathbb{E}_{s \sim \boldsymbol{x}}[u_p(s)])$ that is independent of the value of $\boldsymbol{y}$. To achieve this we can use a trick similar to the one used to make $\mathtt{OPT} = 0$ in Section 3 by augmenting vectors $\boldsymbol{y} \in \mathcal{Y}$ with an extra dimension $\varnothing$ such that $\boldsymbol{y}[\varnothing] = 1$ always holds. Then we have [4]

$$\mathbf{U}_{sj} = \begin{cases} \sum_p u_p(s), & j = \varnothing \\ -\boldsymbol{s}_p[a] u_p(\mathbf{1}_b, \boldsymbol{s}_{-p}), & \text{otherwise} \end{cases}$$

where $\mathbf{1}_b$ denotes the vector having all $0$, apart from index $b$, which is $1$. Note that the number of rows of $\mathbf{U}$ might be doubly-exponential (exponential both in the number of players and the dimension of the polyhedral strategies), which is in contrast to the original Ellipsoid Against Hope algorithm that only allowed a number of rows exponential in the number of players.

**Lemma 4.1.** *Let $G$ be a polyhedral game with pure strategy set $\Pi_p$ for every player $p \in [n]$. Additionally, let $\Phi_p$ be a set of linear transformations for every $p \in [n]$. If $\boldsymbol{x} \in \mathcal{X} = \Delta(\Pi_1 \times \cdots \times \Pi_n)$ and $\boldsymbol{y} = (\phi_1, \ldots, \phi_n) \in \mathcal{Y} = \Phi_1 \times \cdots \times \Phi_n$ then*

$$\boldsymbol{x}^\top \mathbf{U} \boldsymbol{y} = \sum_p \mathbb{E}_{s \sim \boldsymbol{x}}[u_p(s) - u_p(\phi_p(\boldsymbol{s}_p), \boldsymbol{s}_{-p})].$$

---

[4] We are slightly abusing the notation here and use $u_p(\mathbf{1}_b, \boldsymbol{s}_{-p})$ instead of $\mathbf{1}_b \cdot \nabla u_p(\boldsymbol{s}_{-p})$.

It is now evident that our goal is to compute a joint distribution that is a solution to the following linear program: find $\boldsymbol{x} \in \mathcal{X}$ s.t. $\min_{\boldsymbol{y} \in \mathcal{Y}} \boldsymbol{x}^\top \mathbf{U} \boldsymbol{y} \geq 0$.

Observe that this is slightly different from the required non-negativity in Definition 2.6; there we want the individual (per-player) expectations to be non-negative, while here it suffices for the minimum of the *sum* of expectations to be non-negative. However, we can assume without loss of generality that the identity transformation is always a valid transformation[5]. Then, every LP solution $\boldsymbol{x} \in \mathcal{X}$ will satisfy $\boldsymbol{x}^\top \mathbf{U} \boldsymbol{y} \geq 0$ for all $\boldsymbol{y} \in \mathcal{Y}$, including $\boldsymbol{y} = (\phi_1, \mathbf{I}, \ldots, \mathbf{I}), (\mathbf{I}, \phi_2, \ldots, \mathbf{I}), \ldots$ that correspond to to the individual expectations.

The previous LP respects exactly the structure of $(P)$ that our min-max framework can handle. The only remaining component to get a polynomial-time algorithm is to have an efficient good-enough-response oracle GER. Specifically, for any valid $\boldsymbol{y} \in \mathcal{Y}$, we need to respond with an $\boldsymbol{x}$ such that $\boldsymbol{x}^\top \mathbf{U} \boldsymbol{y} \geq 0$. The important insight that allows us to construct an efficient oracle and uncover sparse solutions is that we can always find such an $\boldsymbol{x}$ that is a product distribution — similar to the original Ellipsoid Against Hope algorithm (Papadimitriou and Roughgarden, 2008) that was based on the observation by Hart and Schmeidler (1989). Note that we can always represent a product distribution by simply specifying its marginals and those, in turn, can always be represented as polyhedral strategies. Thus, representing the product distribution $\boldsymbol{x}$ only requires linear space in the game size.

Next, we define an important property that a game must have to enable the efficient implementation of the GER oracle we propose (Lemma D.1) for our algorithm.

**Assumption 4.2** (Polynomial utility gradient property). *Given a product distribution $\boldsymbol{x} \in \Delta(\Pi_1 \times \cdots \times \Pi_n)$, it is possible to compute the value of*

$$\boldsymbol{g}_p(\boldsymbol{x}_{-p}) = \mathop{\mathbb{E}}_{\boldsymbol{s}_{-p} \sim \boldsymbol{x}_{-p}} [\nabla u_p(\boldsymbol{s}_{-p})]$$

*for all players $p \in [n]$ in polynomial time in the encoding length of $\boldsymbol{x}$ and the size of the game.*

This assumption generalizes the polynomial expectation property defined in Papadimitriou and Roughgarden (2008) to more general, polyhedral games. In particular, if we have a normal-form game, the polynomial expectation property amounts to computing $\boldsymbol{g}_p(\boldsymbol{x}_{-p}) \cdot \boldsymbol{x}_p$ for a product distribution $\boldsymbol{x}$. Moreover, as we stated in the introduction, this assumption is very natural for one more reason; it is implicitly assumed in every no-regret learning algorithm.

**Remark 4.3.** *Papadimitriou and Roughgarden (2008) also defined a second property that is required for efficient computation, called the "polynomial type" property. Even though our algorithm does* not *require this property, a variant of it is implicit in the fact that the complexity of the algorithm depends polynomially in the number of players and the dimension of every player's strategy set $\mathcal{A}_p$. However, this relaxation is what allows our algorithm to handle much broader classes of games, such as the extensive-form games that do not have the polynomial type property.*

**Theorem 4.4.** *Let $G$ be a polyhedral game (Definition 2.4) of $n$ players and $\{\Phi_p\}$ be a collection of polytopes corresponding to sets of linear strategy transformations that map every strategy set $\mathcal{A}_p$ to itself. Additionally, let $N = \sum_p d_p^2$. Assume that*

- *there exist polynomial-time separation oracles for $\mathcal{A}_p$ and $\Phi_p$,*

- *$G$ satisfies the polynomial utility gradient property (Assumption 4.2),*

- *$\psi$ is an upper bound on the facet-complexity of every $\mathcal{A}_p$ and $\Phi_p$,*

- *$\log u$ is the maximum encoding length of the utilities of $G$.*

*Then there exists an algorithm that computes an exact $\{\Phi_p\}$-equilibrium of $G$ in time $\mathrm{poly}(N, \log u, \psi)$ and performs $\mathrm{poly}(N, \log u, \psi)$ number of calls to all the separation oracles. Additionally, the equilibrium is represented as a convex combination of at most $N$ pure strategy profiles.*

As a first application of this framework, we argue that it can be applied to normal-form games that satisfy the polynomial type and the polynomial expectation property, defined in Papadimitriou and Roughgarden (2008).

---

[5]Otherwise we can replace each $\Phi_p$ with $\mathrm{co}\{\Phi_p \cup \{\mathbf{I}\}\}$ which remains a rational polytope and admits a separation oracle when $\Phi_p$ has a separation oracle.

**Corollary 4.5** (Exact CE in normal-form games)**.** *If a normal-form game $G$ has the polynomial type and the polynomial expectation property, defined in Papadimitriou and Roughgarden (2008), then our algorithm computes an exact correlated equilibrium of $G$ and runs in polynomial time in the size of the game.*

As we have discussed, a very notable example of polyhedral games is that of extensive-form games. Next, we apply Theorem 4.4 to this class of games, and specifically to the set of all linear-swap deviations, recently defined in Farina and Pipis (2023). In particular, this set contains all trigger deviations and thus, our algorithm also produces an extensive-form correlated equilibrium (EFCE) in a conceptually simpler manner than in the early work of Huang and von Stengel (2008).

**Corollary 4.6** (Exact LCE computation)**.** *There exists an algorithm that runs in $\mathrm{poly}(N, \log u)$ time and computes an exact linear-deviation correlated equilibrium (LCE) in an extensive-form game.*

Finally, we prove in Theorem 4.7 that, at least in the case of computing $\Phi$-equilibria in polyhedral games, the use of a GER over a best-response oracle is not just more elegant, but it is also necessary because constructing a best-response oracle is NP-hard. At the heart of our hardness result lies a reduction from SAT to equilibrium computation in extensive-form games that has also been used in the past to prove the hardness of equilibrium selection for EFCE and LCE (von Stengel and Forges, 2008; Farina and Pipis, 2023). In a sense, constructing a best-response oracle is as hard as the equilibrium selection problem, while constructing a good-enough-response oracle amounts to computing fixed-points of strategy transformation functions. This further highlights the importance of having a framework akin to the one presented in Section 3 for designing new algorithms; the hardness result rules out solutions that require responses competitive against *any* threshold, but sometimes it is sufficient to only compete with a particular good-enough threshold.

**Theorem 4.7** (Hardness of BR oracle)**.** *It is NP-hard to construct a best-response oracle for the Correlator in the Correlator-Deviator game.*

## 5 Discussion and Future Work

In this paper, we devise a polynomial-time algorithm for computing min-max equilibria in bilinear zero-sum games, by utilizing a good-enough-response oracle. We use this machinery to develop a simple general framework for the *exact* computation of $\Phi$-equilibria in polyhedral games for sets $\Phi$ of linear strategy transformations. This framework parallels that of Gordon et al. (2008) on no-regret dynamics, but for exact equilibrium computation. Applying this to extensive-form games, we construct the first polynomial-time algorithm for computing exact linear-deviation correlated equilibria in extensive-form games – a question that had been left open by Farina and Pipis (2023).

We believe that having a simple framework to use as a mental model to guide algorithm design is of paramount importance for the advancement of the field. The $\Phi$-regret minimization framework of Gordon et al. (2008) is indicative of this fact, because it has been key to many interesting results over the years (Morrill et al., 2021; Farina et al., 2022a; Anagnostides et al., 2022; Farina and Pipis, 2023). Compared to no-regret learning, the problem of exact equilibrium computation has been much less studied (basically only in Papadimitriou and Roughgarden (2008); Jiang and Leyton-Brown (2015); Huang and von Stengel (2008)) and we hope that offering a simplified framework will give new insights to advance this front, perhaps aiding in the discovery of new, more practical, algorithms.

Several key questions remain underinvestigated.

- Despite its great theoretical importance, our framework (based on the ellipsoid algorithm) has a polynomial time complexity of rather large degree. Could one devise a more practical alternative while retaining a similar level of generality?
- Can our framework be easily generalized to convex strategy spaces (instead of polytopes)?
- Is there a similar algorithmic framework to compute exact $\Phi$-equilibria in extensive-form games for non-linear transformations $\Phi$? In recent work, Zhang et al. (2024a) give parameterized algorithms for minimizing $\Phi$-regret when $\Phi$ is the set of all degree-$k$ polynomial swap deviations. Can similar guarantees be achieved for high-precision computation of these $\Phi$-equilibria?
- Can ideas similar to those presented in this paper be applied to Markov games?

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

## A   Related work

Algorithms for computing equilibria can be classified broadly into three categories:

- *Polynomial-time algorithms* compute an *exact* equilibrium in time polynomial in the input game size. Note that exact equilibria only make sense when the game has rational utilities, otherwise we settle for $\epsilon$-approximate equilibria in time polynomial in $\log(1/\epsilon)$ and the size of the game.
- *Fully polynomial-time approximation schemes (FPTASs)* compute $\epsilon$-approximate equilibria in time polynomial in $1/\epsilon$ and the size of the input game.
- *Polynomial-time approximation schemes (PTASs)* compute $\epsilon$-approximate equilibria in time that is polynomial in the size of the input game, for every fixed $\epsilon > 0$—however, they might in general have an exponential dependence on $1/\epsilon$.

Nash equilibria are known to be PPAD-complete, thus ruling out any polynomial-time algorithm for them (Daskalakis et al., 2009). Additionally, approximating a Nash equilibrium, even for a constant $\epsilon$ is known to also be PPAD-complete for $n$-player games (Rubinstein, 2015) and to require quasi-polynomial time for 2-player games, assuming ETH for PPAD (Rubinstein, 2016), thus ruling out any PTAS or FPTAS algorithm.

The complexity landscape is significantly more favorable for the case of correlated equilibrium (CE). Specifically, Hart and Mas-Colell (2000); Blum and Mansour (2007) gave efficient no-regret

dynamics (minimizing the so-called *internal regret*) that, if used by all players in a game, can be used to compute an $\epsilon$-approximate CE in normal-form games in time polynomial in the size of the game and $1/\epsilon$. This constitutes the first FPTAS for the computation of CEs. Finally, Papadimitriou and Roughgarden (2008) gave a centralized algorithm that exactly computes a CE in a concisely represented normal-form game in polynomial time in the size of the game. This was the first polynomial-time algorithm for CEs.

The complexity of CE is not settled for extensive-form games (EFGs), and its determination remains a major unresolved question in the field (von Stengel and Forges, 2008; Farina and Pipis, 2023). An advance in this direction was provided very recently by the breakthrough results from two concurrent works of Dagan et al. (2024) and Peng and Rubinstein (2024), which provide a PTAS for CE in extensive-form games, though it remains unclear whether a polynomial-time algorithm or even an FPTAS exist.

Due to the conjectured intractability of computing a normal-form CE in EFGs, researchers have come up with other notions of equilibrium (von Stengel and Forges, 2008; Morrill et al., 2021; Farina and Pipis, 2023; Zhang et al., 2024b) that lie on a spectrum of $\Phi$-equilibria. $\Phi$-equilibria provide a generalization of CE that ranges from least hindsight-rational (coarse correlated equilibria), to maximum hindsight rational (CE), depending on the size of the set of behavior transformation $\Phi$ considered by the players. One of the most notable and natural notions of sequential rationality is that of the extensive-form correlated equilibrium (EFCE) (von Stengel and Forges, 2008). The EFCE was shown to be efficiently computable exactly (Huang and von Stengel, 2008) using a method similar to the Ellipsoid Against Hope of Papadimitriou and Roughgarden (2008). Additionally, it was shown that there exist efficient regret dynamics (minimizing the so called trigger regret) that can be used to compute an $\epsilon$-approximate equilibrium by Farina et al. (2022a). This also gives an FPTAS for EFCE.

Currently, the highest notion of rationality that admits an FPTAS is the recently defined linear-deviation correlated equilibrium (LCE) (Farina and Pipis, 2023), which subsumes previous equilibrium notions such as EFCE. Farina and Pipis (2023) proved that there exist efficient regret dynamics (minimizing linear-swap regret) that can converge to an LCE; the polynomial complexity was then later improved by Zhang et al. (2024b). They however left open the question as to whether there exists a polynomial time algorithm that can compute an exact LCE. We resolve this open question in this paper, showing that the exact computability of equilibria in EFGs extends up to the linear-deviation correlated equilibrium.

**Relationship with work on combinatorially-structured games.** Our algorithmic framework in Section 3 can be used to compute exact min-max equilibria in bilinear games when one of the players has an exponentially large action space and we can only use a good-enough-response oracle—a weaker notion than the best-response oracle.

In light of applications related to Machine Learning and Deep Learning, there has recently been increased renewed interest in games having exponential (or even infinite) action spaces. For example, Assos et al. (2023) propose regret dynamics that can converge to approximate coarse correlated equilibria in infinite (nonparametric) games when a suitable notion of dimension of the game (Littlestone and fat-threshold) is bounded. Dagan et al. (2024) generalize this result even further proving that there also exist regret-dynamics in these cases that can converge in approximate correlated equilibria. Interestingly, recent breakthroughs in Large Language Models have inspired work on "language-based" games that typically have an enormous number of strategies (FAIR et al., 2022; Gemp et al., 2024).

However, the interest in games involving large strategy spaces is by no means a recent phenomenon. For instance, the community of security games has traditionally been interested in the problem of computing Stackelberg equilibria for games where one player (the leader) can possibly have exponentially many strategies (Kiekintveld et al., 2009; Xu, 2016). Another notable example is that of the Colonel Blotto game, which involves exponentially large strategy sets in both players (Ahmadinejad et al., 2019). Finally, work on *learning* in combinatorially-structured games has found applications to online optimization on combinatorial domains such as EFG strategy spaces and flow polytopes (Farina et al., 2022b; Koolen et al., 2010; Takimoto and Warmuth, 2003).

Even though our framework for computing min-max equilibria might have some resemblance to some of the methods in these papers, to the best of our knowledge, in all of the past cases the algorithms cleverly exploit the special combinatorial structure inherent in security games and usually involve

reducing the dimensionality of the strategy space as a first step. Our framework on the other hand, might allow for applications where the large decision set of a player is not amenable to some kind of smaller-dimensional representation.

Finally, we remark that an interesting recurring theme in games with large strategy spaces is that they often assume some kind of best-response oracle access.

**Relationship with work based on best-response oracle access to games** Using best-response oracles is a ubiquitous technique for learning in games or equilibrium computation, starting from the foundational method of fictitious play (Brown, 1951) where players apply a best-response oracle at every round to respond to the empirical frequency of play of their opponent. Best-response oracles (or variants thereof) have additionally been used in security games (Ahmadinejad et al., 2019; Xu et al., 2014), in bilinear games (Gidel et al., 2017), in efficient learning on polytopes (Chakrabarti et al., 2024), in the computation of well-supported equilibria in bilinear games (Goldberg and Marmolejo-Cossío, 2021), in the PSRO for Reinforcement Learning (Lanctot et al., 2017), in infinite games (Assos et al., 2023; Dagan et al., 2024). We remark however that in this paper we do *not* use a best-response oracle in our algorithms. Rather, we use a weaker notion that we coin "Good-Enough-Response" (GER) oracle. In certain cases (such as the computation of $\Phi$-equilibria in Section 4) it is critical to relax the requirement for a best-response oracle because no such oracle can be constructed unless P = NP (Theorem 4.7).

## B  Further preliminaries on linear programs

Here, we introduce some more specialized Lemmas that will be useful for proving the main results of Section 3. The first one concerns feasibility sets of the form we use in our main LPs and shows that they are, in fact, polytopes and that their facet-complexity is properly bounded.

**Lemma B.1.** *Let* $\mathbf{A}$ *be a matrix and*

$$\mathcal{P} = \left\{ \boldsymbol{u} \in \mathcal{U} \;\middle|\; \max_{\boldsymbol{q} \in \mathcal{Q}} \boldsymbol{q}^\top \mathbf{A} \boldsymbol{u} \le c \right\}.$$

*If the following conditions hold*

- $\mathcal{U}$ *be a rational polyhedron with facet-complexity at most* $\varphi$,

- $\mathcal{Q}$ *be the convex hull of a set of finitely many points* $V(\mathcal{Q}) = \{\hat{\boldsymbol{q}}_1, \ldots, \hat{\boldsymbol{q}}_K\}$,

- *the inequality* $(\hat{\boldsymbol{q}}^\top \mathbf{A}) \boldsymbol{u} \le c$ *has encoding length at most* $\varphi$ *for all vertices* $\hat{\boldsymbol{q}} \in V(\mathcal{Q})$.

*Then the set* $\mathcal{P}$ *is a rational polytope with facet-complexity at most* $\varphi$.

*Proof.* First we show that $\mathcal{P} = \mathcal{P}'$, where

$$\mathcal{P}' := \left\{ \boldsymbol{u} \in \mathcal{U} \;\middle|\; (\hat{\boldsymbol{q}}^\top \mathbf{A}) \boldsymbol{u} \le c \; \forall \hat{\boldsymbol{q}} \in V(\mathcal{Q}) \right\},$$

is the polytope defined by finitely many inequality constraints, corresponding to the vertices of $\mathcal{Q}$. Combining this with the assumption that each such inequality has encoding length at most $\varphi$, the result follows immediately. It remains to prove the desired set equality:

- Case $\mathcal{P} \subseteq \mathcal{P}'$:
$$\boldsymbol{u} \in \mathcal{P} \implies \boldsymbol{u} \in \mathcal{U}, \; \max_{\boldsymbol{q} \in \mathcal{Q}} \boldsymbol{q}^\top \mathbf{A} \boldsymbol{u} \le c \implies \boldsymbol{u} \in \mathcal{U}, \; (\hat{\boldsymbol{q}}^\top \mathbf{A}) \boldsymbol{u} \le c \; \forall \hat{\boldsymbol{q}} \in V(\mathcal{Q}),$$

  where the last implication follows from the fact that for all $\hat{\boldsymbol{q}} \in V(\mathcal{Q}) \subseteq \mathcal{Q}$,
$$(\hat{\boldsymbol{q}}^\top \mathbf{A}) \boldsymbol{u} \le \max_{\boldsymbol{q} \in \mathcal{Q}} \boldsymbol{q}^\top \mathbf{A} \boldsymbol{u} \le c.$$

- Case $\mathcal{P} \supseteq \mathcal{P}'$: By definition, any point $\boldsymbol{q} \in \mathcal{Q}$ can be written as the convex combination of all vertices $\boldsymbol{q} = \sum_i^K \lambda_i \hat{\boldsymbol{q}}_i$. Thus, we have
$$\boldsymbol{u} \in \mathcal{P}' \implies \boldsymbol{u} \in \mathcal{U}, \; (\hat{\boldsymbol{q}}^\top \mathbf{A}) \boldsymbol{u} \le c \; \forall \hat{\boldsymbol{q}} \in V(\mathcal{Q}) \implies \boldsymbol{u} \in \mathcal{U}, \; \max_{\boldsymbol{q} \in \mathcal{Q}} \boldsymbol{q}^\top \mathbf{A} \boldsymbol{u} \le c,$$

where the last implication holds because for any $q \in \mathcal{Q}$,

$$q^\top \mathbf{A} u = \sum_i^K \lambda_i \hat{q}_i^\top \mathbf{A} u \leq \sum_i^K \lambda_i c.$$

This concludes the proof. $\qquad\qquad\qquad\qquad\qquad\qquad\qquad\qquad\qquad\qquad\qquad\quad$ $\square$

We also mention a result in the spirit of the classical Farkas lemma. We use this result in to prove our main Theorem of Section 3.

**Lemma B.2** (Generalized Farkas lemma). *Let $\mathcal{X} \subset \mathbb{R}^M, \mathcal{Y} \subset \mathbb{R}^N$ be convex compact sets. Then exactly one of the following two statements is true.*

*1. There exists $x \in \mathcal{X}$ such that $\min_{y \in \mathcal{Y}} x^\top \mathbf{A} y \geq 0$.*

*2. There exists $y \in \mathbb{R}_+ \mathcal{Y}$ such that $\max_{x \in \mathcal{X}} x^\top \mathbf{A} y \leq -1$.*

*Proof.* I) We first show that (1) and (2) cannot be true simultaneously. Assume otherwise and let $\hat{x} \in \mathcal{X}, \hat{y} \in \mathbb{R}_+ \mathcal{Y}$ be values that satisfy (1) and (2) respectively. Since $\hat{y}$ belongs to the conic hull of $\mathcal{Y}$ it must be $\hat{y} = ky'$ for some $k > 0$ and $y' \in \mathcal{Y}$. Thus,

$$\max_{x \in \mathcal{X}} x^\top \mathbf{A} \hat{y} \leq -1 \implies \max_{x \in \mathcal{X}} x^\top \mathbf{A} y' \leq -\frac{1}{k}.$$

Additionally, it holds

$$0 \leq \min_{y \in \mathcal{Y}} \hat{x}^\top \mathbf{A} y \leq \hat{x}^\top \mathbf{A} y' \leq \max_{x \in \mathcal{X}} x^\top \mathbf{A} y' \leq -\frac{1}{k},$$

which is a contradiction. Thus, the statements (1) and (2) cannot be true simultaneously.

II) We now proceed to prove that when (2) is false then (1) must be true. We begin by showing that (2) being false implies that for any $\gamma > 0$ there does not exist any $y \in \mathbb{R}_+ \mathcal{Y}$ such that $\max_{x \in \mathcal{X}} x^\top \mathbf{A} y \leq -\gamma$. Suppose otherwise; then $y' = y/\gamma$ is a multiple of an element of $\mathbb{R}_+ \mathcal{Y}$ and thus $y' \in \mathbb{R}_+ \mathcal{Y}$. Furthermore,

$$\max_{x \in \mathcal{X}} x^\top \mathbf{A} y' = \frac{1}{\gamma} \max_{x \in \mathcal{X}} x^\top \mathbf{A} y \leq -1,$$

which is a contradiction because we have assumed that (2) is false. It directly follows that

$$\min_{y \in \mathcal{Y}} \max_{x \in \mathcal{X}} x^\top \mathbf{A} y \geq 0.$$

By the minimax theorem it also holds

$$\max_{x \in \mathcal{X}} \min_{y \in \mathcal{Y}} x^\top \mathbf{A} y \geq 0 \implies \exists_{x \in \mathcal{X}} : \min_{y \in \mathcal{Y}} x^\top \mathbf{A} y \geq 0$$

and thus, statement (1) is true.

III) Finally, we need to prove the inverse direction; when (1) is false then (2) must be true. This is trivial because if (2) was false, then by II) we would have that (1) is true, which contradicts the assumption. $\qquad\qquad\qquad\qquad\qquad\qquad\qquad\qquad\qquad\qquad\qquad\qquad\qquad\qquad\qquad$ $\square$

## C  Omitted proofs from Section 3

**Theorem 3.1.** *If the following hold*

*1. $\mathcal{X} \subset \mathbb{R}^M, \mathcal{Y} \subset \mathbb{R}^N$ are rational polytopes and $\mathcal{Y}$ has facet-complexity at most $\varphi$,*

*2. we have access to a separation oracle $\mathtt{SEP}_\mathcal{Y}$ for $\mathcal{Y}$ and a good-enough-response oracle $\mathtt{GER}$,*

*3. the encoding length of $x^\top \mathbf{A}$ is at most $\varphi$ for all $\mathtt{GER}$ oracle responses and all vertices of $\mathcal{X}$,*

---
**Algorithm 2:** Separation oracle for the ellipsoid method on $(D)$
---
**Input:** Separation oracle $\text{SEP}_{\mathbb{R}_+\mathcal{Y}}$ for $\mathbb{R}_+\mathcal{Y}$, and a good-enough-response oracle GER.
**Output:** A separating hyperplane $\boldsymbol{c}$ for $\boldsymbol{y}$ in $(D)$, and a corresponding vector $\boldsymbol{x}$ from GER, if it
        exists.
**if** $\text{SEP}_{\mathbb{R}_+\mathcal{Y}}$ *deems that* $\boldsymbol{y}$ *is in* $\mathbb{R}_+\mathcal{Y}$ **then**
    |   Set $(\boldsymbol{x}, \boldsymbol{c})$ to the output $(\boldsymbol{x}, \mathbf{A}^\top\boldsymbol{x})$ of GER($\boldsymbol{y}$);
**else**
    |   Set $\boldsymbol{c}$ to the separating hyperplane output by $\text{SEP}_{\mathbb{R}_+\mathcal{Y}}$;
    |   $\boldsymbol{x} = \varnothing$;
**end**
---

*then Algorithm 1 runs in* $\text{poly}(N, \varphi)$ *time, performs* $L = \text{poly}(N, \varphi)$ *oracle calls, and computes an exact solution* $\boldsymbol{x}^*$ *of* $(P)$ *that is a mixture of at most* $N$ *oracle responses. In particular, the encoding length of* $\boldsymbol{x}^*$ *depends polynomially on the encoding length of the* GER *oracle responses.*

*Proof.* First, we can assume without loss of generality that there exists a dimension $\varnothing$ in all $\boldsymbol{y} \in \mathcal{Y}$ such that $\boldsymbol{y}[\varnothing] = 1$. Otherwise, it is always possible to augment these vectors with an extra dimension before applying the next steps of the algorithm. This allows us to convert the given separation oracle $\text{SEP}_{\mathcal{Y}}$ into a new separation oracle $\text{SEP}_{\mathbb{R}_+\mathcal{Y}}$ for $\mathbb{R}_+\mathcal{Y}$ that we can then use to construct the general oracle of Algorithm 2.

The first step of the algorithm is to execute the ellipsoid method on $(D)$. Using the assumptions of the theorem in Lemma B.1 with $\mathcal{U} = \mathbb{R}_+\mathcal{Y}$ and $\mathcal{Q} = \mathcal{X}$, it follows that $(D)$ is a polytope and has facet-complexity at most $\varphi$. Additionally, by the fact that $(P)$ is feasible (it has an equilibrium with $\text{OPT} = 0$) and by Lemma B.2 it follows that $(D)$ must be infeasible.

To execute the ellipsoid method on $(D)$ would then mean, in the language of Grötschel et al. (1993), to solve the Strong Nonemptiness Problem for $(D)$ using the strong separation oracle of Algorithm 2. To this end, we use the algorithm from Theorem 6.4.1 of Grötschel et al. (1993). This algorithm works for any polyhedron, even if it is not bounded or full-dimensional, as might be the case here. To do that, it might execute the central-cut ellipsoid method more than once, but never more than $N$ times. In our case, we already know that $(D)$ is infeasible and thus, the algorithm terminates after $N$ executions of the central-cut ellipsoid and concludes that $(D)$ is infeasible.

Since the central-cut ellipsoid method is an oracle-polynomial algorithm that is executed $N$ times in polyhedra of facet-complexity at most $\varphi$, the whole process runs in polynomial time and performs a polynomial number of separation oracle calls. To calculate the exact number $L$ of oracle calls, we note that the algorithm in Grötschel et al. (1993, Theorem 6.4.1) initializes the central-cut ellipsoid method with

$$R = 2^{O(N^2\varphi)} \text{ and } \epsilon = 2^{-O(N^5\varphi)},$$

while the central-cut method terminates in $O(N\log(1/\epsilon) + N^2\log R)$ iterations (Grötschel et al., 1993, Theorem 3.2.1). Combining these with the fact that the central-cut ellipsoid method is repeated $N$ times, we get that the number of oracle calls is $L = O(N^7\varphi)$.

Next, note that $(D')$ is comprised of constraints coming from GER oracle responses, which by Lemma B.1 gives that the facet-complexity of $(D')$ must also be at most $\varphi$. By going through the same process as before, the algorithm will reach the same conclusion after executing the central-cut ellipsoid method $N$ times; $(D')$ is infeasible.

Finally, by the infeasibility of $(D')$ and Lemma B.2, it follows that $(P')$ must be feasible. An equivalent way to express $(P')$ is

$$\text{find } \boldsymbol{a}$$
$$\text{s.t. } \min_{\boldsymbol{y} \in \mathcal{Y}} \boldsymbol{a}^\top(\mathbf{X}^\top\mathbf{A})\boldsymbol{y} \geq 0$$
$$\boldsymbol{a} \in \Delta^L,$$

where $\Delta^L$ is the $L$-dimensional simplex and $\mathbf{X} = [\boldsymbol{x}_1 \mid \cdots \mid \boldsymbol{x}_L]$ is a matrix with the GER oracle responses as its columns.

Applying Lemma B.1 for $\mathcal{U} = \Delta^L$ and $\mathcal{Q} = \mathcal{Y}$ we conclude that $(P')$ describes a polytope

$$\mathcal{P} = \left\{ \boldsymbol{a} \in \Delta^L \mid \min_{\boldsymbol{y} \in \mathcal{Y}} \boldsymbol{a}^\top (\mathbf{X}^\top \mathbf{A}) \boldsymbol{y} \geq 0 \right\}$$

of encoding length at most $L \operatorname{poly}(\varphi)$. This is because, for any vertex $\hat{\boldsymbol{y}} \in V(\mathcal{Y})$, the inequality $\boldsymbol{a}^\top (\mathbf{X}^\top \mathbf{A}) \hat{\boldsymbol{y}} \geq 0$ has $L$ coefficients, each of which having encoding length $\operatorname{poly}(\varphi)$. This can be solved in polynomial time by any known linear programming method. Even better, it is possible to compute a basic feasible solution of this LP, which will have at most $N$ non-zero entries and thus the final solution $\boldsymbol{x}^* = \mathbf{X}\boldsymbol{a}$ will be a mixture of at most $N$ oracle responses. $\qquad\square$

## D    Omitted proofs from Section 4

**Lemma 4.1.** *Let $G$ be a polyhedral game with pure strategy set $\Pi_p$ for every player $p \in [n]$. Additionally, let $\Phi_p$ be a set of linear transformations for every $p \in [n]$. If $\boldsymbol{x} \in \mathcal{X} = \Delta(\Pi_1 \times \cdots \times \Pi_n)$ and $\boldsymbol{y} = (\phi_1, \ldots, \phi_n) \in \mathcal{Y} = \Phi_1 \times \cdots \times \Phi_n$ then*

$$\boldsymbol{x}^\top \mathbf{U} \boldsymbol{y} = \sum_p \mathop{\mathbb{E}}_{s \sim \boldsymbol{x}} [u_p(s) - u_p(\phi_p(\boldsymbol{s}_p), \boldsymbol{s}_{-p})].$$

*Proof.* As we discussed, each linear transformation $\phi_p$ can be viewed as a $d_p \times d_p$ transformation matrix $\mathbf{B}_p$. We denote the matrix entries with $\mathbf{B}_p[b, a]$. In particular, if $\boldsymbol{s}'_p = \phi_p(\boldsymbol{s}_p)$, we have $\boldsymbol{s}'_p[b] = \sum_a \mathbf{B}_p[b, a] \boldsymbol{s}_p[a]$. Then for any $\boldsymbol{x} \in \mathcal{X}$ and $\boldsymbol{y} = (\phi_1, \ldots, \phi_n) \in \mathcal{Y}$ we have

$$
\begin{aligned}
\boldsymbol{x}^\top \mathbf{U} \boldsymbol{y} &= \sum_s x_s \sum_p \left( u_p(s) - \sum_{a \in [d_p]} \sum_{b \in [d_p]} \mathbf{B}_p[b, a] \boldsymbol{s}_p[a] u_p(\mathbf{1}_b, \boldsymbol{s}_{-p}) \right) \\
&= \sum_p \sum_s x_s \left( u_p(s) - u_p \left( \sum_{b \in [d_p]} \mathbf{1}_b \sum_{a \in [d_p]} \mathbf{B}_p[b, a] \boldsymbol{s}_p[a], \boldsymbol{s}_{-p} \right) \right) \\
&= \sum_p \sum_s x_s \left( u_p(s) - u_p \left( \sum_{b \in [d_p]} \mathbf{1}_b \boldsymbol{s}'_p[b], \boldsymbol{s}_{-p} \right) \right) \\
&= \sum_p \sum_s x_s \left( u_p(s) - u_p(\phi_p(\boldsymbol{s}_p), \boldsymbol{s}_{-p}) \right) \\
&= \sum_p \mathop{\mathbb{E}}_{s \sim \boldsymbol{x}} [u_p(s) - u_p(\phi_p(\boldsymbol{s}_p), \boldsymbol{s}_{-p})],
\end{aligned}
$$

where in the second equality we have used the multi-linearity of the utilities. $\qquad\square$

Now we are ready to present the good-enough-response oracle that will allow us to develop an efficient algorithm for computing exact $\Phi$-equilibria in polyhedral games. As a general backbone, this Lemma follows the constructive proof that Papadimitriou and Roughgarden (2008) did for the CE existence result of Hart and Schmeidler (1989) and, crucially, it produces pure strategies using the idea of Jiang and Leyton-Brown (2015).

**Lemma D.1** (GER oracle for $\Phi$-equilibria). *For every $\boldsymbol{y} \in \mathcal{Y} = \Phi_1 \times \cdots \times \Phi_n$ there exists a pure strategy profile $s \in \Pi_1 \times \cdots \times \Pi_n$ such that $\mathbf{1}_s^\top \mathbf{U} \boldsymbol{y} \geq 0$. Furthermore, such a strategy profile alongside with the vector $\mathbf{1}_s^\top \mathbf{U}$ can be computed efficiently, provided that the game satisfies the polynomial utility gradient property (Assumption 4.2) and there exists a polynomial-time separation oracle for every $\mathcal{A}_p$.*

*Proof.* As we have discussed, we can denote all linear transformations $\phi_p$ using a matrix $\mathbf{B}_p$ such that $\phi_p(\boldsymbol{x}_p) = \mathbf{B}_p \boldsymbol{x}_p$.

First note that there always exists a fixed-point of any linear strategy transformation $\phi_p$; this follows from Brouwer's fixed-point theorem and the fact that these transformations are continuous maps of a

**Algorithm 3:** Purified `GER` oracle

---

**Input:** Polyhedral game G of $n$ players, $\boldsymbol{y} \in \mathcal{Y} = \Phi_1 \times \cdots \times \Phi_n$, separation oracles $\text{SEP}_{\mathcal{A}_p}$ for all $p \in [n]$.

**Output:** A pure strategy profile $s \in \Pi_1 \times \cdots \times \Pi_n$ such that $\boldsymbol{1}_s^\top \mathbf{U} \boldsymbol{y} \geq 0$.

Compute a product distribution $\boldsymbol{x}$ s.t. $\boldsymbol{x}^\top \mathbf{U} \boldsymbol{y} = 0$ by finding fixed points $\boldsymbol{x}_p = \phi_p(\boldsymbol{x}_p)$ for all $p \in [n]$ ;

**for** $p \in [n]$ **do**

    Find a set of $k \leq d_p + 1$ vertices $\left\{ \hat{\boldsymbol{s}}_p^{(1)}, \ldots, \hat{\boldsymbol{s}}_p^{(k)} \right\} \subset V(\mathcal{A}_p)$ s.t. $\boldsymbol{x}_p = \sum_{i=1}^{k} \lambda_i \hat{\boldsymbol{s}}_p^{(i)}$ ;

    Set $\boldsymbol{s}_p^*$ to the vertex $\hat{\boldsymbol{s}}_p^{(i)}$ that satisfies $\boldsymbol{x}_{(p \to \boldsymbol{s}_p^*)}^\top \mathbf{U} \boldsymbol{y} \geq 0$;

    Set $\boldsymbol{x}$ to be $\boldsymbol{x}_{(p \to \boldsymbol{s}_p^*)}$ ;

**end**

Finally, $\boldsymbol{x}$ must correspond to a pure strategy profile $s$;

---

compact convex set $\mathcal{A}_p$ to itself. Additionally, since the transformations are linear we can always efficiently compute a fixed-point of any transformation by solving the following LP:

$$\text{find } \boldsymbol{x}_p$$
$$\text{s.t. } \mathbf{B}_p \boldsymbol{x}_p = \boldsymbol{x}_p$$
$$\boldsymbol{x}_p \in \mathcal{A}_p$$

that can be solved in polynomial time using the ellipsoid method with the given separation oracle for $\mathcal{A}_p$.

Next, let us restrict our attention only to product distributions $\boldsymbol{x} \in \Delta(\Pi_1 \times \cdots \times \Pi_n)$. In this case it will be $x_s = x_{-p}(\boldsymbol{s}_{-p}) x_p(\boldsymbol{s}_p)$ for all pure strategy profiles $s$, which gives

$$
\begin{aligned}
\boldsymbol{x}^\top \mathbf{U} \boldsymbol{y} &= \sum_p \sum_{\boldsymbol{s}_{-p}} \sum_{\boldsymbol{s}_p \in \Pi_p} x_{-p}(\boldsymbol{s}_{-p}) x_p(\boldsymbol{s}_p) \left[ u_p(\boldsymbol{s}_p, \boldsymbol{s}_{-p}) - u_p(\phi_p(\boldsymbol{s}_p), \boldsymbol{s}_{-p}) \right] \\
&= \sum_p \sum_{\boldsymbol{s}_{-p}} x_{-p}(\boldsymbol{s}_{-p}) u_p \left( [\boldsymbol{x}_p - \phi_p(\boldsymbol{x}_p)], \boldsymbol{s}_{-p} \right) \\
&= \sum_p \boldsymbol{g}_p(\boldsymbol{x}_{-p}) \cdot [\boldsymbol{x}_p - \phi_p(\boldsymbol{x}_p)],
\end{aligned}
\tag{1}
$$

where $\boldsymbol{x}_p \in \mathcal{A}_p$ is the marginal distribution for player $p$ represented as a point of the polyhedral strategy set. In the second equality we have used the multi-linearity of $u_p(\cdot, \boldsymbol{s}_{-p})$ and the linearity of the transformations; $\sum_{\boldsymbol{s}_p} x_p(\boldsymbol{s}_p) \phi_p(\boldsymbol{s}_p) = \phi_p(\boldsymbol{x}_p)$. It directly follows from the last equality that if we set each marginal distribution equal to the corresponding fixed-point $\boldsymbol{x}_p = \phi_p(\boldsymbol{x}_p)$, we get a product distribution $\boldsymbol{x}$ such that $\boldsymbol{x}^\top \mathbf{U} \boldsymbol{y} = 0$.

Now, it remains to find a way to extract the desired pure strategy profile $s$ from this product distribution. We follow a similar procedure to the purification technique used by Jiang and Leyton-Brown (2015). Similar to their algorithm, we define $\boldsymbol{x}_{(p \to \boldsymbol{s}_p)}$ for a product distribution $\boldsymbol{x}$ to be the product distribution in which player $p$ plays pure action $\boldsymbol{s}_p \in \mathcal{A}_p$ and all other players act according to $\boldsymbol{x}_{-p}$. Additionally, note that since $\mathcal{A}_p$ is a polytope, it must hold

$$\boldsymbol{x}_p = \sum_{\boldsymbol{s}_p \in V(\mathcal{A}_p)} \lambda_{\boldsymbol{s}_p} \boldsymbol{s}_p$$

for some convex combination $\{ \lambda_{\boldsymbol{s}_p} \geq 0 \mid \sum_{\boldsymbol{s}_p} \lambda_{\boldsymbol{s}_p} = 1 \}$. By the product distribution structure, it is easy to see that for every player $p \in [n]$,

$$\boldsymbol{x}^\top \mathbf{U} \boldsymbol{y} = \sum_{\boldsymbol{s}_p \in V(\mathcal{A}_p)} \left[ \boldsymbol{x}_{(p \to \boldsymbol{s}_p)}^\top \mathbf{U} \boldsymbol{y} \right] \lambda_{\boldsymbol{s}_p} \tag{2}$$

The algorithm of Jiang and Leyton-Brown (2015) iterates over all players and for each player $p$ they search over all its pure strategies and find one, $\boldsymbol{s}_p^*$, for which $\boldsymbol{x}_{(p \to \boldsymbol{s}_p^*)}^\top \mathbf{U} \boldsymbol{y} \geq 0$. Such a pure

strategy must always exist because (2) represents a convex combination over all vertices (a.k.a. pure strategies). However, in our case we cannot iterate over all pure strategies for a player because they might be exponentially many.

To make this procedure general for all polyhedral games, we observe that by Carathéodory's theorem there must always exist a subset $\left\{\hat{s}_p^{(1)}, \ldots, \hat{s}_p^{(k)}\right\} \subset V(\mathcal{A}_p)$ of at most $k \leq d_p + 1$ vertices of $\mathcal{A}_p$ that satisfy

$$\boldsymbol{x}_p = \sum_{i=1}^{k} \lambda_i \hat{\boldsymbol{s}}_p^{(i)}$$

for some convex combination represented with $\lambda_1, \ldots, \lambda_k$. Thus, we can follow the same procedure as before but this time only search over $k$ vertices instead of all (possibly exponentially many) vertices of $\mathcal{A}_p$. The complete algorithm is shown in Algorithm 3.

This can be implemented in polynomial time because: (a) there exists an algorithmic version of Carathéodory's theorem Grötschel et al. (1993, Theorem 6.5.11) that only requires access to a separation oracle for $\mathcal{A}_p$, and (b) Assumption 4.2 allows us to compute $\boldsymbol{x}_{(p \to \boldsymbol{s}_p^*)}^\top \mathbf{U} \boldsymbol{y}$ in polynomial time for any product distribution $\boldsymbol{x}_{(p \to \boldsymbol{s}_p^*)}$, as is evident from (1). $\qquad\square$

**Theorem 4.4.** *Let $G$ be a polyhedral game (Definition 2.4) of $n$ players and $\{\Phi_p\}$ be a collection of polytopes corresponding to sets of linear strategy transformations that map every strategy set $\mathcal{A}_p$ to itself. Additionally, let $N = \sum_p d_p^2$. Assume that*

- *there exist polynomial-time separation oracles for $\mathcal{A}_p$ and $\Phi_p$,*

- *$G$ satisfies the polynomial utility gradient property (Assumption 4.2),*

- *$\psi$ is an upper bound on the facet-complexity of every $\mathcal{A}_p$ and $\Phi_p$,*

- *$\log u$ is the maximum encoding length of the utilities of $G$.*

*Then there exists an algorithm that computes an exact $\{\Phi_p\}$-equilibrium of $G$ in time $\mathrm{poly}(N, \log u, \psi)$ and performs $\mathrm{poly}(N, \log u, \psi)$ number of calls to all the separation oracles. Additionally, the equilibrium is represented as a convex combination of at most $N$ pure strategy profiles.*

*Proof.* The set $\mathcal{X} = \Delta(\Pi_1 \times \cdots \times \Pi_n)$ is trivially a rational polytope and the set $\mathcal{Y} = \Phi_1 \times \cdots \times \Phi_n \subset \mathbb{R}^N$ is the Cartesian product of rational polytopes, hence a rational polytope. Furthermore, we can directly construct a polynomial-time separation oracle for $\mathcal{Y}$ by calling the separation oracles for each one of the sets $\Phi_p$. Additionally, every row of the $\mathbf{U}$ matrix has $N$ entries, each with encoding length at most $2 \log u$. Using the good-enough-response oracle from Lemma D.1, each response $(\boldsymbol{x}_k, \boldsymbol{x}_k^\top \mathbf{U}) \in \mathcal{X} \times \mathbb{Q}^N$ corresponds to a pair of a pure strategy profile (vertex of $\mathcal{X}$) and a row of $\mathbf{U}$. Thus, each response has encoding length at most $2N \log u$. Set $\varphi = \max(2N \log u, \psi)$.

Now, we can apply Theorem 3.1, which gives us an algorithm running in $\mathrm{poly}(N, \varphi)$ time and performing $\mathrm{poly}(N, \varphi)$ oracle calls. Combining this with Lemma D.1, it follows that the total time complexity is $\mathrm{poly}(N, \varphi) = \mathrm{poly}(N, \log u, \psi)$. Finally, the optimal solution $\boldsymbol{x}^*$ will be comprised of a mixture of $N$ oracle responses. In other words, $\boldsymbol{x}^*$ will be an exact $\{\Phi_p\}$-equilibrium for the game $G$ with probability mass on at most $N$ pure strategy profiles. $\qquad\square$

**Corollary 4.5** (Exact CE in normal-form games)**.** *If a normal-form game $G$ has the polynomial type and the polynomial expectation property, defined in Papadimitriou and Roughgarden (2008), then our algorithm computes an exact correlated equilibrium of $G$ and runs in polynomial time in the size of the game.*

*Proof.* A normal-form game is a polyhedral game where every strategy set is a probability simplex. Additionally, the set $\Phi$ of all linear transformations in normal-form games is that of swap-deviations which is equivalent to the set of all stochastic matrices (Gordon et al., 2008). Both the sets of strategies and the sets of stochastic matrices can easily be represented as polytopes of bounded facet-complexity having polynomially many constraints. Finally, the polynomial utility gradient property

(Assumption 4.2) reduces to the polynomial expectation property in succinct normal-form games and the polynomial type property is implicitly satisfied (see Remark 4.3). Thus, all requirements of Theorem 4.4 are satisfied and we conclude that there exists a polynomial time algorithm for computing CEs in normal-form games. □

**Corollary 4.6** (Exact LCE computation). *There exists an algorithm that runs in* $\mathrm{poly}(N, \log u)$ *time and computes an exact linear-deviation correlated equilibrium (LCE) in an extensive-form game.*

*Proof.* We apply Theorem 4.4 for the set of linear-swap deviations. Specifically, in Farina and Pipis (2023, Theorem 3.1) it is proved that:

- The set $\Phi_{\mathrm{LIN}}$ of linear-swap deviations for a player $p$ is a rational polytope.

- This polytope can be described using a polynomial number of equality constraints, which immediately implies the existence of an efficient separation oracle.

- Every constraint of the characterization has at most $|\Sigma_p|^2$ coefficients, each belonging to $\{0, 1, -1\}$. Thus, the facet-complexity of $\Phi_{\mathrm{LIN}}$ must be $\psi = |\Sigma_p|^2$.

Finally, since the number of non-zero utilities are at most equal to the game tree size, it trivially follows that extensive-form games satisfy the polynomial utility gradient property (Assumption 4.2). It follows that there exists a polynomial time algorithm for computing LCEs. □

**Theorem 4.7** (Hardness of BR oracle). *It is NP-hard to construct a best-response oracle for the Correlator in the Correlator-Deviator game.*

*Proof.* A best-response oracle for the Correlator in the Correlator-Deviator game must respond with the optimal $\boldsymbol{x} \in \mathcal{X} = \Delta(\Pi_1 \times \cdots \times \Pi_n)$ for any given $\boldsymbol{y} \in \mathcal{Y} = \Phi_1 \times \cdots \times \Phi_n$. More precisely, we have to be able to compute

$$\boldsymbol{x}^* = \arg\max_{\boldsymbol{x} \in \mathcal{X}} \left\{ \boldsymbol{x}^\top \mathbf{U} \boldsymbol{y} \right\} = \arg\max_{\boldsymbol{x} \in \mathcal{X}} \left\{ \sum_p \mathbb{E}_{s \sim \boldsymbol{x}} [u_p(s) - u_p(\phi_p(\boldsymbol{s}_p), \boldsymbol{s}_{-p})] \right\}$$

for all $\boldsymbol{y} \in \mathcal{Y}$.

To prove that this process is intractable it suffices to find a game and an equilibrium concept such that it is NP-hard to compute $\boldsymbol{x}^*$ for at least one $\boldsymbol{y} \in \mathcal{Y}$. For the solution concept we choose the coarse-correlated equilibrium, in which the sets $\Phi_p$ consist of all constant (or external) deviations that output a fixed strategy $\phi_p(\boldsymbol{x}_p) = \bar{\boldsymbol{s}}_p \in \Pi_p$ no matter the input strategy $\boldsymbol{x}_p$. For the game, we choose the SAT-game that was also used to prove the hardness of equilibrium selection for EFCE and LCE in extensive-form games (von Stengel and Forges, 2008; Farina and Pipis, 2023). The exact details are not important for our purposes, but we will only use the fact that any SAT instance can be encoded in a 2-player extensive-form game using a polynomial-time reduction. In this game, any pure strategy profile $s \in \Pi_1 \times \cdots \times \Pi_n$ with social welfare (sum of players' utilities) equal to 2 corresponds to a satisfying assignment for the SAT instance, while any other strategy profile has social welfare at most $2(1 - 1/n)$. Thus, there exists a Nash equilibrium (and hence, a CCE) corresponding to a pure strategy profile that has maximum social welfare.

Before we proceed, we augment the SAT-game by adding an extra decision point for both players at the beginning of the game that asks whether they want to play. If both players respond "Yes" then the game continues as normal, otherwise –if at least one player responds "No"– the game ends and both players get a 0 payoff. We denote the pure strategy of the "No" answer from player 1 with $\boldsymbol{s}_1^N$ and the "No" answer from player 2 with $\boldsymbol{s}_2^N$.

Now, to prove the desired result consider a Correlator-Deviator game applied to the computation of a CCE in the above augmented SAT-game. Assume that there exists a polynomial-time best-response oracle for this game that returns a solution of polynomial size in the representation of the game (and hence, the size of the SAT instance). Then, we can use it to best-respond to $\boldsymbol{y} = (\phi_1, \phi_2)$ for

$\phi_1(\boldsymbol{x}_1) = \boldsymbol{s}_1^N$ and $\phi_2(\boldsymbol{x}_2) = \boldsymbol{s}_2^N$. Specifically, we have

$$\boldsymbol{x}^* = \arg\max_{\boldsymbol{x}\in\mathcal{X}}\left\{\sum_p \mathbb{E}_{s\sim\boldsymbol{x}}[u_p(s) - u_p(\phi_p(\boldsymbol{s}_p), \boldsymbol{s}_{-p})]\right\}$$

$$= \arg\max_{\boldsymbol{x}\in\mathcal{X}}\left\{\sum_p \mathbb{E}_{s\sim\boldsymbol{x}}[u_p(s)] - \sum_p \mathbb{E}_{s\sim\boldsymbol{x}}[u_p(\boldsymbol{s}_p^N, \boldsymbol{s}_{-p})]\right\}$$

$$= \arg\max_{\boldsymbol{x}\in\mathcal{X}}\left\{\sum_p \mathbb{E}_{s\sim\boldsymbol{x}}[u_p(s)]\right\}.$$

In other words, the BR oracle returns in this case a distribution $\boldsymbol{x}^*$ over pure strategy profiles with maximum social welfare. Since the BR oracle computes a polynomially-sized $\boldsymbol{x}^*$ in polynomial-time, we can uncover a pure strategy profile of maximum social welfare that corresponds, by construction of the SAT-game, to a satisfying assignment of the SAT instance. We conclude that constructing a best-response oracle in the Correlator-Deviator game corresponding to the compution of a CCE in extensive-form games is NP-hard. □

