# OpenReview forum: "Polynomial-Time Computation of Exact $\Phi$-Equilibria in Polyhedral Games"
_NeurIPS.cc/2024/Conference — NeurIPS 2024 spotlight_

### Official Review · Reviewer_un2H · 2024-06-25

**Soundness:** 3
**Presentation:** 4
**Contribution:** 3
**Rating:** 7
**Confidence:** 3

**Summary:**

The authors this generalize the Ellipsoid Against Hope and develop a simple algorithmic framework for efficiently computing saddle-points in bilinear zero-sum games, even when one of the dimensions is exponentially large.

**Strengths:**

The main strength of this work is the significance of the problem considered.

**Weaknesses:**

See questions below.

**Questions:**

Page 1:
Can you please give some more details about multi-player games?

Page 2:
I do not understand Lines 52 -- 58. What do you mean by uncoupled learning dynamics?

Page 3:
Can you please elaborate on the technicalities behind Item 2?

Page 4:
Can you please explain a bit more the definition of $\Phi$-equilibrium?

Page 5:
Line 218:
How do you exactly find the scaling factor?

Page 6:
I got confused by ($D'$).

How do you combine the GER responses to represent $x^*$?

Page 7:
Line 302:
I do not understand this math display.

Page 8:
Assumption 4.2 needs to be motivated better :)

Page 9:
Can you please elaborate on the future work, especially on the bullet regarding Markov games? Thank you!

**Limitations:**

None.

---

> ### Author Rebuttal · Authors · 2024-08-05
>
> Thank you for the positive review and for helping us improve the presentation of our paper. We will incorporate your suggestions in the final version. We respond to your questions below:
>
> * Page 1: Here, with "multi-player games" we refer to games with any number $n$ of players.
>
> * Page 2: Uncoupled learning dynamics refer to no-regret learning dynamics that can be followed by all players in a decentralized manner. The example equilibria that we give in these lines constitute equilibria that can be approximated with known efficient no-regret dynamics.
>
> * Page 3: The requirement for linear transformations $\phi \in \Phi$ implies that $\mathbb{E}[\phi(x)] = \phi(\mathbb{E}[x])$. For example, this property is used in Lemma 4.1, which is a crucial component of our characterization of exact $\Phi$-equilibrium computation as a bilinear zero-sum game.
>
> * Page 4: Intuitively, a $\{ \Phi_p \}$-equilibrium is a distribution $\mu$ over the joint strategy profiles such that, on expectation, no player $p$ has an incentive to unilaterally deviate from the recommended joint strategy $\mathbf{s} \sim \mu$ using any transformation $\phi \in \Phi_p$. Please see Line 181 for a formal definition. We will also incorporate this intuitive description in the final version of our paper.
>
> * Page 5: Since $\mathbf{y}' = \alpha \mathbf{y}$ it should hold that $\mathbf{y}'[\emptyset] = \alpha$. Thus, to find the scaling factor $\alpha$ we simply need to look at the value of $\mathbf{y}'$ in the extra dimension.
>
> * Page 6: If we let $x_1, \dots, x_L$ be the GER response vectors, then (D') is basically identical to (D) but replacing $\mathcal{X}$ with the convex hull $\text{co}\{ x_k \}$ of the response vectors. Then, to compute the final feasible solution $x^*$ it suffices to solve the linear program (P'). Any valid solution of that LP belongs to $\text{co}\{ x_k \}$ and thus, it is a convex combination of the GER response vectors.
>
> * Page 7: In this math display, we set the values for the meta-game utility matrix $\mathbf{U}$. Specifically, for the column corresponding to the extra augmented dimension $j = \emptyset$, the value of $U_{s j}$ is the sum of all players' utilities for the current joint strategy profile. In all other columns $j = (p, a, b)$ we have that $U_{s j}$ is the product of $-\mathbf{s}_p[a]$ (the $a$-th component of player $p$'s strategy vector $\mathbf{s}_p$) and $ u_p(  \mathbf{1}\_b , s\_{ -p } ) $ (the utility of player $p$ in strategy profile $s$ under the strategy vector $\mathbf{1}_b$).
>
> * Page 8: We believe that this assumption is very natural because (a) in normal-form games, it is equivalent to the natural polynomial expectation property that Papadimitriou and Roughgarden have defined in their Ellipsoid Against Hope algorithm and (b) as we have stated in page 2 of the introduction, it is already implicitly assumed in every no-regret learning algorithm. We will also add this short motivation in page 8 as well, thanks!
>
> * Page 9: The open question related to Markov Games would be to investigate whether it is possible to compute exact equilibria of some kind in Markov Games. Note that this is an important class of games that do not directly fall under the category of polyhedral games, so our framework cannot be applied directly. To the best of our knowledge, the question of computing exact correlated equilibria in Markov Games has not been generally studied in the past.

---

> > ### Comment · Reviewer_un2H · 2024-08-08
> >
> > Thank you! :)

---

### Official Review · Reviewer_r2xR · 2024-07-13

**Soundness:** 3
**Presentation:** 3
**Contribution:** 3
**Rating:** 7
**Confidence:** 3

**Summary:**

This paper gives an algorithm for computing exact $\Phi$-equilibria in polyhedral games. The algorithm follows the general framework of the "Ellipsoid Against Hope" method for computing exact correlated equilibria in succinctly represented normal form games. The paper generalizes this method to computing linear $\Phi$-equilibria (i.e. correlated equilibria where each player's allowed set of deviations are encoded by $\Phi$), in polyhedral games. The set $\Phi$ of allowed individual player deviations is required to consist of linear transformations on the player's strategy set. This model captures previously studied variants of correlated equilibria, most notably linear-deviation correlated equilibria, where $\Phi$ is the set of all linear-swap deviations on sequence-form strategies. Thus the results in this paper give the first polynomial time algorithm for exact computation of linear-deviation correlated equilibria in extensive-form games.

The key modification needed to the "Ellipsoid Against Hope" method is to allow for the use of something weaker than a best-response oracle as a subroutine in solving the two-player meta-game induced by the original polyhedral game. The authors show that for exact linear $\Phi$-equilibrium computation, it is NP-hard to compute a best-response oracle, but it is tractable to compute a "good enough"-response oracle. The method then needs to be modified to use the "good-enough"-response oracle to efficiently compute an exact equilibrium. The notion of "good-enough" response here means that the response need only achieve utility equal to that at equilibrium in the two-player meta-game.

**Strengths:**

- The key insight that only a good-enough response oracle is sufficient for using the "ellipsoid against hope" method is simple yet powerful. It gives a more straightforward analysis of prior results on exact computation of correlated equilibria, and solves an open problem on exact computation of linear-swap correlated equilibria.

- The proof that it is NP-hard to compute a best-response oracle demonstrates the necessity of using good-enough response oracles as introduced in this paper, and thus suggests that these techniques could be useful for exact computation of equilibria in other settings.

- The paper is well-written and easy to follow.

**Weaknesses:**

- A possible weakness is that most of the techniques in the paper (including those that yield the good-enough response oracle) are straight-forward modifications of the "ellipsoid against hope" method. I don't necessarily view this as much of a weakness, because the primary insight of the paper is that one does not actually need a best response oracle for this method, and that good-enough response oracles can be constructed using standard techniques.

- Another possible weakness is in the specificity of the problem studied: exact computation of linear-swap correlated equilibria in polyhedral games. Equilibrium computation results are known for approximate equilibria, for exact equilibria in normal form games, and for extensive form correlated equilibria (which are a special case of linear-swap correlated equilibria).

**Questions:**

Is there an example of game where exact computation of a linear-$\Phi$ equilibrium has a natural interpretation that is somehow better than an approximate equilibrium, and where $\Phi$ naturally should consist of linear functions?

**Limitations:**

Yes.

---

> ### Author Rebuttal · Authors · 2024-08-05
>
> Thank you for your review and observations on our paper. We respond to the weaknesses and questions you raised.
>
> * > [Weakness] straightforward modifications of the ellipsoid against hope method
> >
> Our framework balances generality with simplicity, as it greatly simplifies the algorithm of [1] for EFCE, while at the same time providing the first polynomial-time algorithm for computing the very general linear correlated equilibria in extensive-form games [2] (thus, going beyond the standard "polynomial type" property of the ellipsoid against hope).\
> Despite the apparent simplicity of our framework, we do agree that it offers insightful ideas for algorithm design, including the good-enough-response oracles. Finally, as we argue in the Discussion section, we believe that offering simplified algorithmic frameworks can greatly aid algorithm development in the general area.
>
>
> * > [Weakness] specificity of the problem studied
> >
> Our paper gives the first polynomial-time algorithm for computing exact linear-deviation correlated equilibria in extensive-form games, thus capturing the largest yet known set of equilibria in these games that can be computed exactly. Regarding exact equilibria, our work subsumes all previous results in normal-form games, as well as the exact computation of EFCE for extensive-form games (with EFCE being a strictly weaker solution concept than linear equilibria). Finally, we give sufficient conditions that can allow for computing exact linear equilibria in any general polyhedral game.
>
> * > [Question] Is there an example of game where exact computation of a linear-Phi equilibrium has a natural interpretation that is somehow better than an approximate equilibrium, and where
> Phi naturally should consist of linear functions?
> >
> In this paper we show for the first time that we can compute linear correlated equilibria in polyhedral games in polynomial time. While our focus is exact equilibria, if we stop the ellipsoid iteration earlier, we can also compute $\epsilon$-approximate equilibria in $\text{polylog}(1 / \epsilon)$ steps. This is exponentially better than no-regret dynamics that, instead, require $\text{poly}(1/\epsilon)$ steps. Thus, in theory, it is possible to compute equilibria in much higher precision than by using no-regret dynamics.
>
> ### References
>
> [1] Wan Huang and Bernhard von Stengel. 2008. Computing an extensive-form correlated equilibrium in polynomial time. In International Workshop on Internet and Network Economics
>
> [2] Gabriele Farina and Charilaos Pipis. 2023. Polynomial-Time Linear-Swap Regret Minimization in Imperfect-Information Sequential Games. In Thirty-seventh Conference on Neural Information Processing Systems.

---

> > ### Comment · Reviewer_r2xR · 2024-08-13
> >
> > Thanks for the response! I have read your rebuttal and the other reviews and will increase my score to 7.

---

### Official Review · Reviewer_2L7d · 2024-07-13

**Soundness:** 4
**Presentation:** 4
**Contribution:** 2
**Rating:** 7
**Confidence:** 4

**Summary:**

This paper studies the problem of computing phi-equilibria in a general class of games called polyhedral games. Phi-equilibria are a class of game-theoretic equilibria where each player has low regret with respect to some class Phi of linear transformation functions (e.g. this captures various notions of correlated equilibria). Polyhedral games are games where each player has a convex polyhedral action set and where the payoffs are given by multilinear functions (also capturing cases of interest such as normal-form games, Bayesian games, extensive-form games, ...).

In 2008, Papadimitriou and Roughgarden introduced the Ellipsoid Against Hope (EAH) algorithm which showed how to efficiently compute succinctly representable correlated equilibria in some classes of multiplayer games (including games where writing down a generic correlated equilibrium might take exponential time). This paper generalizes this algorithm to the above much broader set of equilibria and games. In particular, the authors introduce an algorithm which efficiently constructs a Phi-equilibrium in an n-player polyhedral game, with the only requirement being that the class Phi of linear transformation functions has an efficient separation oracle (along with some other mild constraints).

As with the EAH algorithm, the main technical ingredient is to show that there exists such a correlated equilibrium which can be expressed as a convex combination of a small number of product distributions. To do this, the authors generalize the original technique used in the EAH paper to the problem of computing equilibria in two-player bilinear zero-sum games. In particular, they show how to efficiently compute a minimax equilibrium for one player (in time polynomial in the dimension of their action set) even if the other player has an extremely high-dimensional action set. The idea here is that as long as you can find “good enough responses” to actions of the other player (responses that guarantee you the minimax equilibrium), you can use this find a low dimensional subspace of the opponent’s high-dimensional action space in which a minimax equilibrium must be supported, and finally solve the resulting problem explicitly.

**Strengths:**

Evaluation

Understanding the computational complexity of computing various types of game-theoretic equilibria is one of the fundamental questions in the area of learning in games. Recently, there has a been a surge of interest in understanding how to compute various types of correlated equilibria in general-sum games (e.g. recent improvements to the best known swap regret bounds). This paper fits into this line of work, answering a question of the complexity of computing exact forms of correlated equilbria in “large” games (e.g. extensive-form games) by providing a very natural set of conditions for when this is possible.

In some sense, it is accurate to describe this paper as a mostly straightforward generalization of the EAH algorithm to these more general settings -- the main technical observation (i.e., the reason why the EAH algorithm does not work “out of the box”) being that for linear swap transformations, the complexity of the solution should only depend on the dimensionality of your action space, not on the number of pure strategies (number of extreme points). But I think this is still a valuable observation, and the consequences of this (that you can exactly compute these equilibria for reasonable classes of phi) are not a priori obvious and would be of interest to NeurIPS researchers in this area.

I also very much enjoyed the presentation of this paper. I think it is nice how this paper abstracts out the fundamental problem that the EAH technique solves (find a minimax strategy in an unbalanced two-player game, given only a “good enough response” oracle); I think this gave me a better understanding of the original EAH algorithm, and I would not be surprised if this rephrasing of the subproblem ends up useful elsewhere.

**Weaknesses:**

See above

**Questions:**

Feel free to reply to any element of the review.

**Limitations:**

Limitations adequately addressed.

---

> ### Author Rebuttal · Authors · 2024-08-05
>
> Thank you for the positive review. You are right that in our generalization of the EAH, it is critical that the time complexity depends on the dimensionality of the action space and not on the number of pure strategies (which might be exponentially many in the dimensionality of the action space, as is the case in extensive-form games). This goes beyond the standard Ellipsoid Against Hope algorithm's requirement for the "polynomial type" property. We believe that the simplicity and generality of our framework for computing exact $\Phi$-equilibria is especially nice, as it greatly simplifies the algorithm of [1] for EFCE, while at the same time providing the first polynomial-time algorithm for computing the very general linear correlated equilibria in extensive-form games [2]. Finally, we completely agree that the framework for computing equilibria in bilinear zero-sum games using GERs should be of independent interest, beyond computing $\Phi$-equilibria in games.
>
> ### References
>
> [1] Wan Huang and Bernhard von Stengel. 2008. Computing an extensive-form correlated equilibrium in polynomial time. In International Workshop on Internet and Network Economics
>
> [2] Gabriele Farina and Charilaos Pipis. 2023. Polynomial-Time Linear-Swap Regret Minimization in Imperfect-Information Sequential Games. In Thirty-seventh Conference on Neural Information Processing Systems.

---

> > ### Comment · Reviewer_2L7d · 2024-08-12
> >
> > Thank you for your response. I have read through the other reviews/responses and maintain my positive evaluation of this paper.

---

### Official Review · Reviewer_Ht5B · 2024-07-21

**Soundness:** 3
**Presentation:** 3
**Contribution:** 3
**Rating:** 6
**Confidence:** 5

**Summary:**

The paper, titled "Polynomial-Time Computation of Exact $\Phi$-Equilibria in Polyhedral Games," proposes a novel algorithmic framework to compute saddle-points in bilinear zero-sum games, particularly when one dimension is exponentially large. This framework extends the Ellipsoid Against Hope algorithm and introduces a "good-enough-response" oracle to compute exact linear ε-equilibria in polyhedral games efficiently. The authors claim that this new approach resolves an open question by providing the first polynomial-time algorithm for computing exact linear-deviation correlated equilibria in extensive-form games.

**Strengths:**

### Strengths

1. **Innovative Framework**: The paper presents a significant extension of the Ellipsoid Against Hope algorithm, applying it to a broader class of games, including extensive-form games.

2. **Polynomial-Time Algorithm**: The proposed algorithm offers a polynomial-time solution for computing exact $\Phi$-equilibria, which is a notable advancement in game theory and computational complexity.

3. **Generality and Simplicity**: The framework is described as conceptually simpler than existing methods while being general enough to handle various types of equilibria, such as correlated equilibria and extensive-form correlated equilibria.

4. **Resolution of Open Problem**: The paper addresses and resolves an open question posed by Farina and Pipis (2023) regarding the polynomial-time computation of linear-deviation correlated equilibria.

5. **Technical Depth**: The theoretical foundations and technical depth are robust, with clear definitions, assumptions, and a well-structured algorithmic framework.

**Weaknesses:**

### Weaknesses

1. **Complexity and Practicality**: While the algorithm is polynomial-time, the degree of the polynomial may be high, potentially limiting practical applicability for very large games.

2. **Empirical Validation**: The paper lacks empirical validation or experimental results demonstrating the practical performance of the proposed algorithm on real-world or benchmark datasets.

3. **Assumptions and Limitations**: The framework relies on specific assumptions, such as the polynomial utility gradient property, which may not hold for all types of games or real-world scenarios.

4. **Clarity and Accessibility**: The paper is highly technical, which might make it less accessible to a broader audience without a strong background in game theory and computational complexity.

5. **Comparative Analysis**: There is limited discussion on how the proposed method compares with other state-of-the-art algorithms in terms of computational efficiency and accuracy.

**Questions:**

The paper makes a substantial theoretical contribution to the field of game theory and computational algorithms. Given its innovative approach and resolution of a significant open problem, it is a strong candidate for acceptance at NeurIPS. However, the authors should consider including empirical results and a more detailed comparative analysis to strengthen the practical relevance and impact of their work.

Personally, I have tried multiple times to parse again and again section 4 to understand how GER does really work efficiently. Could you please explain more the reduction to the Correlator-Deviator game.
Additionally, can you explain how you go from summation inequality of lemma 4.1 to the individual parts' inequality. To be honest, I need elicit explanations for each paragraph between lines 297 and 326, which are the crucial part to make the algorithm works.

What is the limit to apply non-linear transformation?
In order to get a Nash equilibrium, which kind of transformation family do you really need?

Additionally, improving the clarity and accessibility of the paper could broaden its appeal and understanding among the NeurIPS community.

---

> ### Author Rebuttal · Authors · 2024-08-05
>
> Thank you for your thorough review and your comments on the paper. We respond below to the raised weaknesses and questions.
>
> * > [Weakness] Complexity and Practicality
> >
> Indeed, the degree of the polynomial in our algorithm's time complexity is extremely high, rendering it impractical. However, it is a first step in the direction of understanding what classes of equilibria can be computed exactly and efficiently. Currently, in the literature there are no exact-equilibrium algorithms with practical applicability that applies to large classes of games. Our hope is that our simplified algorithmic framework will offer an avenue for future research that will lead to other, more practical such algorithms.
>
> * > [Weakness] Empirical Validation
> >
> The main contribution of our paper is theoretical; we propose the first polynomial-time algorithm for computing exact $\Phi$-equilibria in polyhedral games (including extensive-form games, which was a previous open question). Our algorithm includes several nested executions of the ellipsoid method, which has an unwieldy time complexity and is tricky to implement right in practice. We have thus left this as one of the problems for future work.
>
> * > [Weakness] Assumptions and Limitations
> >
> The polynomial utility gradient assumption is standard and satisfied by many classes of games. Among others, it is satisfied by extensive-form games and by succinct normal-form games [1], which include graphical games, anonymous games, polymatrix games, congestion games, scheduling games, and local effect games. Additionally, as we mention in our paper, this assumption is implicitly made in all no-regret learning algorithms, which are the gold standard for computing equilibria in large games in practice. It is also worth mentioning that if we had no assumption on the access of the game's utilities, the task at hand would be inherently impossible, as any algorithm would have to know all entries of the exponentially-sized utility tensor for the game.
>
> * > [Weakness] Comparative Analysis
> >
> As we mention in Line 130, we include an extensive discussion of related work in Appendix A. The main body only contains the few most directly related papers that concern the exact computation of correlated equilibria in games. Finally, since our algorithm provides the first polynomial-time algorithm to exactly compute several classes of equilibria in games (such as linear-deviation correlated equilibria in extensive-form games), there is no direct benchmark we could compare to for these cases.
>
> * > [Question] Correlator-Deviator game
> >
> This is a two-player zero-sum meta-game we define, in which the strategy spaces are: the space of all joint distributions for the first meta-player (Correlator), and the Cartesian product of all sets of deviations for the second meta-player (Deviator). The matrix giving the Correlator's utility is given in lines 302-303.
>
> * > [Question] can you explain how you go from summation inequality of lemma 4.1 to the individual parts' inequality
> >
> We refer to lines 312-318. In a nutshell, if we assume (without loss of generality) that the identity is always a valid deviation $\mathbf{I} \in \Phi_p$, then a feasible solution $\mathbf{x}$ of the LP in line 311 will guarantee that $\mathbf{x}^\top \mathbf{U} \mathbf{y} \geq 0$ for all $\mathbf{y} \in \mathcal{Y}$. Thus, this inequality will also hold for $\mathbf{y} = (\phi_1, \mathbf{I}, \dots, \mathbf{I}), (\mathbf{I}, \phi_2, \dots, \mathbf{I}), \dots$, which correspond to the individual parts' inequalities.
>
> * > [Question] how GER does really work efficiently
> >
> Intuitively, our algorithm casts the problem of $\Phi$-equilibrium computation as one of computing a min-max solution of the Correlator-Deviator bilinear zero-sum game. A min-max solution for the Correlator in this game corresponds to a valid $\Phi$-equilibrium in the original game. Thus, the GER in our case receives a tuple $\mathbf{y} = (\phi_1, \phi_2, \dots, \phi_n)$ of deviations for all players (aka., a Deviator's strategy) and outputs a good-enough strategy $\mathbf{x} \in \Delta(\Pi_1 \times \dots \times \Pi_n)$ for the Correlator. One of the crucial ideas for the efficient computation of such a strategy is to notice that we can always compute a product distribution that is good-enough, owing to the existence of fixed points for linear transformation functions. Then, combining the probabilistic method with techniques for decomposing a point of a polytope into a small number of vertices (such as the Caratheodory theorem), it is possible to extract a single pure joint strategy from this product distribution. For more details, we refer to Lemma D.1 and Algorithm 3 in Lines 719-760 of the Appendix.
>
> * > [Question] To be honest, I need elicit explanations for each paragraph between lines 297 and 326, which are the crucial part to make the algorithm works.
> >
> Thank you for the recommendation. We will edit the final version of the paper to include the more intuitive explanation we provided for paragraph 312-318 and also slightly improve the language in the rest of these lines.
>
> * > [Question] What is the limit to apply non-linear transformation? In order to get a Nash equilibrium, which kind of transformation family do you really need?
> >
> The Nash equilibrium is basically defined as a coarse-correlated equilibrium (ie., using external deviations) with the additional constraint that the joint distribution is a product distribution. If we want to apply no-regret dynamics for the computation of Nash equilibria, we know that this is generally only possible in two-player zero-sum games, where it suffices to compute any coarse-correlated equilibrium of the game. In general, there do not exist any uncoupled dynamics that guarantee convergence to Nash equilibria.
>
> ### References
>
> [1] Christos H. Papadimitriou and Tim Roughgarden. 2008. Computing Correlated Equilibria in Multi-Player Games. JACM

---

### Decision · Program_Chairs · 2024-09-25

**Decision:**

Accept (spotlight)

**Comment:**

This is a strong submission that gives new algorithms for exact Phi-equilibria in several settings.
I recommend accepting for spotlight.


*******************************************

Two writing comments (or really questions to the authors that I wished I had asked earlier):

1. It would be good to elaborate on your second condition that requires the class \Phi to contain only linear transformations.
E.g. explain in more detail what you mean and give a natural example where it doesn't hold.

1-a. You should change l.96 to say that \Phi contains **only** valid linear transformations.
As written it doesn't really make sense because I can always add the linear transformations to \Phi.

2. Given the NP-hardness result for BR, I find it surprising that finding a response with value greater than 0 is easy. For most NP-hard optimization problems, if you ask whether you can beat some threshold that's also NP-hard. It would be nice to elaborate a bit on why GER is still tractable in some settings.